# Activin-mediated alterations of the fibroblast transcriptome and matrisome control the biomechanical properties of skin wounds

Mateusz S. Wietecha [1,6], Marco Pensalfini [2,6], Michael Cangkrama[1], Bettina Müller[2], Juyoung Jin [1], Jürgen Brinckmann[3,4], Edoardo Mazza[2,5 ✉] & Sabine Werner [1 ✉]

Matrix deposition is essential for wound repair, but when excessive, leads to hypertrophic scars and fibrosis. The factors that control matrix deposition in skin wounds have only partially been identified and the consequences of matrix alterations for the mechanical properties of wounds are largely unknown. Here, we report how a single diffusible factor, activin A, affects the healing process across scales. Bioinformatics analysis of wound fibroblast transcriptome data combined with biochemical and histopathological analyses of wounds and functional in vitro studies identify that activin promotes pro-fibrotic gene expression signatures and processes, including glycoprotein and proteoglycan biosynthesis, collagen deposition, and altered collagen cross-linking. As a consequence, activin strongly reduces the wound and scar deformability, as identified by a non-invasive in vivo method for biomechanical analysis. These results provide mechanistic insight into the roles of activin in wound repair and fibrosis and identify the functional consequences of alterations in the wound matrisome at the biomechanical level.

---

[1] Institute of Molecular Health Sciences, Department of Biology, ETH Zurich, Otto-Stern-Weg 7, 8093 Zurich, Switzerland. [2] Institute for Mechanical Systems, Department of Mechanical and Process Engineering, ETH Zurich, Leonhardstrasse 21, 8092 Zurich, Switzerland. [3] Department of Dermatology, University of Lübeck, 23562 Lübeck, Germany. [4] Institute of Virology and Cell Biology, University of Lübeck, 23562 Lübeck, Germany. [5] EMPA, Swiss Federal Laboratories for Materials Science and Technology, Überlandstrasse 129, 8600 Dübendorf, Switzerland. [6] These authors contributed equally: Mateusz S. Wietecha, Marco Pensalfini. ✉email: mazza@imes.mavt.ethz.ch; sabine.werner@biol.ethz.ch

njury to the skin triggers a well-coordinated repair program that aims to restore the wounded area[1]. It is initiated by blood clotting and rapid mounting of an inflammatory response, followed by migration and proliferation of various cell types. This results in the formation of new tissue, which undergoes maturation during a long remodeling phase[1–3]. Unfortunately, skin defects involving the dermis result in formation of scars, which lack all appendages and have reduced tensile strength and deformability[2,3]. Scar formation can be excessive as seen in hypertrophic scars and keloids[4]. The extent of scarring is coordinated by growth factors and cytokines that control deposition and remodeling of connective tissue[5] as well as by biophysical factors, such as mechanical tension[6]. However, the contribution of individual factors to different aspects of the scarring response is largely unknown.

The major producers of the extracellular matrix (ECM) in the wound and the resulting scar are fibroblasts and in particular myofibroblasts[7]. Wound (myo)fibroblasts have recently been well characterized with regard to origin, gene expression profile, cell fate, and scarring potential[8–14]. However, the effect of different biochemical and biophysical factors on the fibroblast expression profile and phenotype in skin remains to be determined. In particular, it is unclear how alterations in the fibroblast transcriptome affect the matrisome and ultimately the mechanical properties of the skin. Addressing this question requires innovative strategies to measure different biomechanical parameters of the wound tissue at physiological levels of tension and stretch. Most published studies on wound biomechanics determined rupture properties of healing tissue, thereby applying largely over-physiological deformations[15,16]. We recently established an ex vivo protocol to characterize wound stiffness at physiological stretch magnitude[17]. So far, the only in vivo investigation providing a local characterization of the mechanical behavior of skin wounds was based on indentation, which pushes the skin downwards. This provided insight into the in vivo level of out-of-plane stiffness of healing tissue, which showed reduced compliance compared to the healthy tissue[18]. Here, we establish a protocol for the non-invasive analysis of physiological in-plane wound deformation behavior in vivo, and we use it to investigate the relationship between the fibroblast transcriptome, the wound matrisome, and the biomechanical properties of wounds over the time course of healing.

We further determine how these parameters are affected by a cytokine that is relevant for the wound healing process. We chose activin A (hereafter activin), since inhibition of its activity by overexpression of its secreted antagonist follistatin in keratinocytes of transgenic mice delayed healing[19], while activin overexpression in keratinocytes accelerated re-epithelialization and also granulation tissue formation of excisional wounds[20,21]. However, activin does not directly enhance keratinocyte proliferation[22], suggesting that it promotes wound healing mainly via cells in the dermis/granulation tissue. Keratinocyte-derived activin can reach these cells because of its high diffusibility. Thus, activin acts as an endocrine acting hormone[23,24] and also as a classical morphogen[25], and activin serum levels are elevated in the transgenic mice overexpressing this protein in keratinocytes[26].

Consistent with an effect of activin on the dermis/granulation tissue, depletion of regulatory T cells reduced the healing-promoting effect of activin[27]. However, activin also promotes proliferation and migration of cultured fibroblasts and expression of collagen type I by these cells[21,28,29], suggesting that it may promote healing via this cell type. Therefore, we use mice over-expressing activin in keratinocytes[21] as a model system to determine the effect of a single cytokine on the wound healing process across scales and we determine the underlying mechanisms in skin fibroblasts in vitro. The results provide mechanistic insight into the effect of activin on wound repair and demonstrate how modifications in the fibroblast transcriptome translate into histological, biochemical, and mechanical alterations of skin wounds.

## Results

**Activin promotes collagen maturation in skin wounds.** To assess potential effects of activin on wound fibroblasts and matrix deposition by these cells, we analyzed Herovici-stained sections of full-thickness wounds. As expected, mice overexpressing activin in keratinocytes (Act mice) exhibited a significantly higher proportion of closed (re-epithelialized) wounds and a larger granulation tissue at day 5 after injury compared to wild-type (WT) controls (Fig. 1a, b). Five- and 7-day wounds of Act mice had a higher density of newly deposited (young) collagen (light blue; Fig. 1a, c, Supplementary Fig. 1a). From day 10 onwards, highly crosslinked, mature collagen (purple) appeared in the wound bed and further extended during the remodeling phase. Wounds from Act mice showed a greater density of mature collagen at these time points and also a longer granulation tissue/early scar tissue at days 10 and 21 post-injury (Fig. 1a, c, d, Supplementary Fig. 1a).

Picrosirius Red staining identified a higher proportion of thick collagen fibers in 5-day wound edges of Act mice (red; Fig. 1e i and ii, f, Supplementary Fig. 1b, c) and a higher abundance of all collagen fibers in the wound centers (Fig. 1e iii and iv, f, Supplementary Fig. 1b, c). Collagen type III was restricted to the wound edge in 5-day wounds of WT mice, but already covered the whole granulation tissue in Act mice. By day 7, it extended to the entire wound bed in mice of both genotypes, but Act wounds exhibited thicker and more densely packed collagen fibers (Supplementary Fig. 1d, e).

**Wound fibroblasts have a distinct transcriptional signature.** To gain insight into the molecular mechanisms underlying the effect of activin on the wound matrix, we characterized fibroblasts of normal and wounded skin. Flow cytometry analysis of wound cell suspensions using the pan-fibroblast marker PDGFRα (CD140a[30]), combined with exclusion of immune cells (Fig. 2a), showed that the relative fibroblast frequency in 3- and 5-day wounds nearly tripled compared to unwounded skin in WT animals and was further elevated (approximately 1.5-fold) in Act mice at these time points, while this difference was no longer observed at day 7 when the wounds were closed (Fig. 2b). Fibroblasts were then FACS-sorted from normal skin (NS) and 5-day wounds (5dw) and subjected to RNA sequencing. Principal component analysis (PCA) of the data showed clear differences between 5dw and NS fibroblast transcriptomes (Supplementary Fig. 2a). The majority of the most highly expressed genes, including the genes for decorin (Dcn) and fibrillar collagens (Col1a1, Col1a2, Col3a1), were expressed at similarly high levels in fibroblasts from NS and 5dw. They enriched for genes expressed in fibroblast cell lines (Fig. 2c, bold, Supplementary Fig. 2b), reflecting the purity of the sorted cells. Wounding induced strong transcriptional alterations in fibroblasts from WT and Act mice, while comparisons of fibroblasts between genotypes yielded only very few highly statistically significant differences in NS or 5dw (Fig. 2d, Supplementary Fig. 2c). PCA of only the 5dw fibroblasts showed a separation between cells from WT and Act mice particularly along PC2 and PC3, although in some cases the inter-sample variability was as high as that between genotypes, probably due to the inherent heterogeneity of the sorted wound fibroblasts (Supplementary Fig. 2d). The apparently weak effect of activin is most likely underestimated, since we used

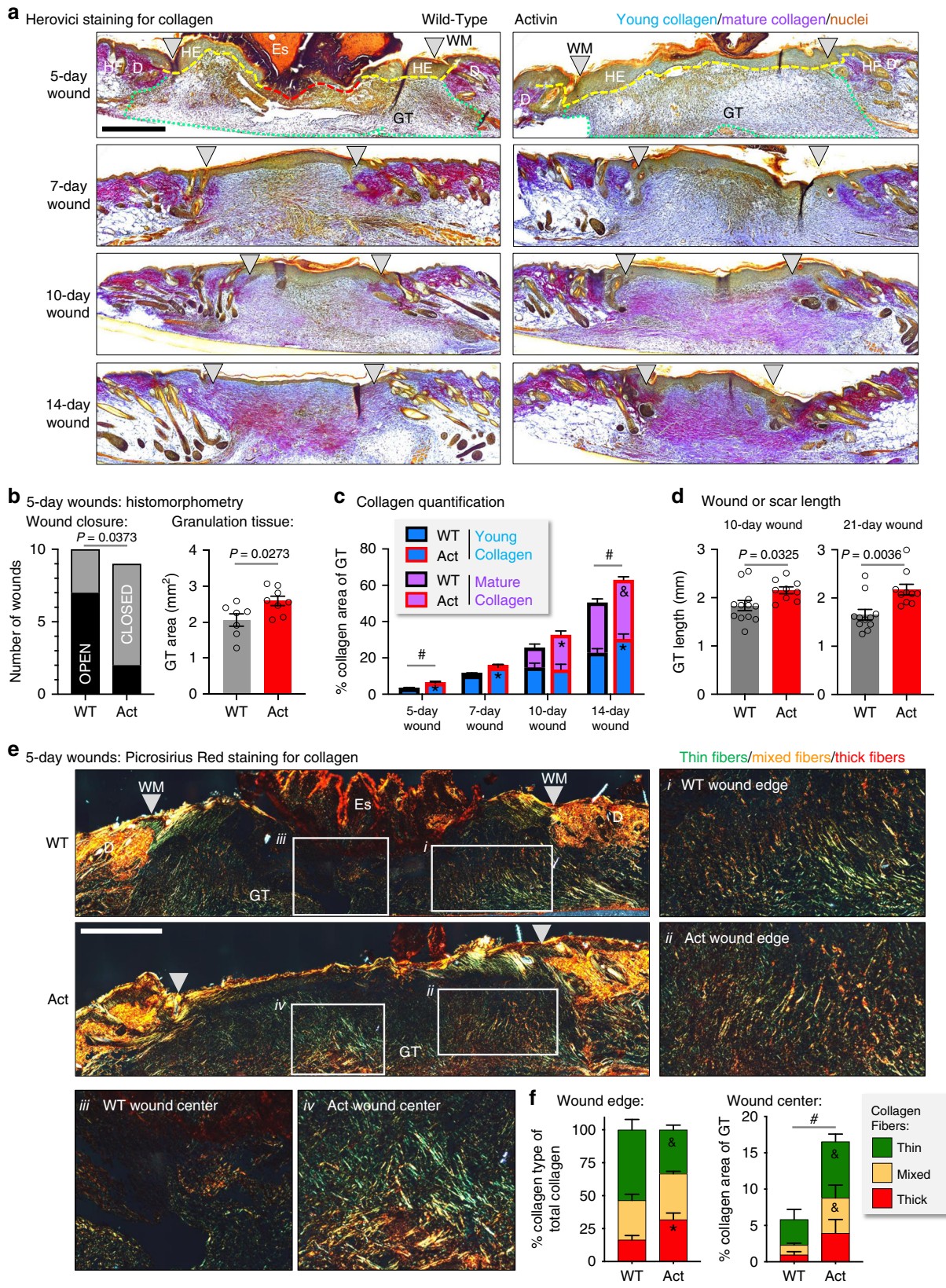

all skin/wound fibroblasts for the analysis. However, the secretion of activin by keratinocytes results in a gradient of activin within the dermis/granulation tissue, with fibroblasts of the deeper dermis/granulation tissue being exposed to lower concentrations. The top activin-regulated genes in NS and 5dw, which still reached statistical significance, were also expressed at biologically relevant levels. They include murine *Inhba* itself (Supplementary Fig. 2e, f), reflecting the previously described activin autoregulation[28], while other activin genes and also activin receptor genes were not regulated (Supplementary Fig. 2c, right). Importantly,

**Fig. 1 Activin promotes collagen deposition and maturation in healing skin wounds. a** Representative photomicrographs of Herovici-stained sections of 5-, 7-, 10-, and 14-day wounds from WT and Act mice. Yellow dotted line represents extent of hyperproliferative wound epidermis; red dotted line represents extent of open wound; green dotted line represents area of granulation tissue. **b** Left: Number of closed (fully re-epithelialized) or open wounds. $n = 10$ for WT, $n = 9$ for Act mice. Right: Granulation tissue area. $n = 7$ for WT, $n = 8$ for Act mice. **c** Quantification of young, mature, and total collagen density in skin wounds. $n = 7, 11, 12, 11$ for WT, $n = 6, 11, 10, 8$ for Act mice, for 5-, 7-, 10-, or 14-day wounds, respectively. $^{\&}P < 0.10$, $^{*}P < 0.05$, $^{\#}P < 0.05$ (total collagen) for Act vs WT at each time point. **d** Length of granulation tissue/scar tissue in 10- and 21-day wounds. $n = 12$ for WT, $n = 9$ for Act mice (10d), and $n = 11$ for WT, $n = 10$ for Act mice (21d). **e** Representative photomicrographs of Picrosirius Red-stained sections from 5-day wounds with insets for corresponding areas in wound edge (i, ii) and wound center (iii, iv). **f** Quantification of collagen fiber types at the wound edges relative to total collagen (left) and in the wound centers relative to GT area (right). $n = 4$ for both genotypes. $^{\&}P < 0.10$, $^{*}P < 0.05$, $^{\#}P < 0.05$ (total collagen) for Act vs WT. Graphs show mean ± SEM and $P$ values; two-sided Chi-square test (**b** left), two-tailed Student's $t$-test (**b** right, **c**, **d**, **f**); see Supplementary Fig. 1a, b for all individual comparisons and $P$ values in **c** and **f**. All $n$ numbers indicate biological replicates. Gray triangles represent wound margins (WM); HE hyperproliferative epithelium, HF hair follicle, D dermis, Es eschar, GT granulation tissue. Scale bars: 500 µm. Source data are provided as a Source Data file (Fig. 1).

several matrix genes, such as those encoding asporin (*Aspn*) and periostin (*Postn*), were strongly regulated by activin in 5dw (Supplementary Fig. 2f).

The majority of statistically significant differentially regulated genes were shared between all 5dw vs NS comparisons (Fig. 2e). The shared up-regulated genes enriched in Gene Ontology (GO) biological process terms for ECM and collagen organization, inflammation, hypoxia response, and angiogenesis (Fig. 2f; Supplementary Fig. 2g for down-regulated genes). Ingenuity Pathway Analysis (IPA) additionally predicted activation of connective tissue cell adhesion, movement, proliferation and adhesion of ECM (Fig. 2g).

Gene Set Enrichment Analysis (GSEA) showed enrichment of our wound fibroblast signature from CD-1/C57BL/6 F1 mixed-background mice for genes highly expressed in myofibroblasts from 7-day small excisional wounds vs NS fibroblasts of C57BL/6 mice[11], myofibroblasts from 12- vs 26-day large excisional wounds of mixed-background mice[10], and all three myofibroblast sub-types from 5-day small excisional wounds of C57BL/6 mice[13] (Fig. 2h). Leading edge analysis showed that myofibroblast marker genes, e.g. the α-smooth muscle actin (α-SMA) gene (*Acta2*), were co-enriched among all wound (myo)fibroblast datasets (Supplementary Fig. 2h). These results point to a remarkable reproducibility of the response of fibroblasts to wounding that is independent of genetic background and wound size.

**Activin induces a pro-fibrotic transcriptome.** Since fibroblasts directly respond to activin[28], we investigated differences in the extent of wounding-induced gene activation in the presence of the *INHBA* transgene, despite minor overall transcriptomic differences between 5dw fibroblasts from Act and WT mice. When comparing relative increases in gene expression (at least 5% higher) in 5dw of Act vs WT mice using NS of Act or WT mice as a baseline, we found a large overlap in relatively up-regulated ECM-related genes, regardless of the genotype of NS (Supplementary Fig. 3a). We then filtered the lists for ECM-encoding genes according to the Matrisome database[31] and found that comparison to the WT NS baseline yielded a higher number of matrix-associated genes in Act 5dw fibroblasts (Supplementary Fig. 3a). Since a comparison to the WT condition detects genes, which are not influenced by activin accumulation prior to wounding, we chose this comparison for further analysis. Importantly, the ECM genes, which showed higher absolute expression in Act vs WT 5dw (Supplementary Fig. 2f), were present in the overlap of the two baseline comparisons (Supplementary Fig. 3a, bold).

We next ranked the genes showing statistically significant regulation by the ratio of log fold changes in 5dw(Act) vs NS (WT) over 5dw(WT) vs NS(WT) comparisons (Supplementary

Fig. 3b). The ranked list contained genes, which showed higher absolute expression in Act vs WT 5dw (Supplementary Fig. 3b, bold), but had little overlap with the most highly expressed genes in the fibroblast transcriptomes (Supplementary Fig. 3c), suggesting that this signature was not dependent on the slightly higher fibroblast density in Act wounds. We then used the ranked list of 165 activin-regulated genes to run comparative GSEA analyses against relevant gene sets from published studies, in comparison to the total wound fibroblast signature. First, we generated a gene set of significantly activin co-expressed genes in multi-tissue datasets via the Search-based Exploration of Expression Compendium (SEEK) database[32]. These genes showed enrichment for ECM-related pathways (Supplementary Fig. 3d, left). GSEA showed strong enrichment of this gene set in our wound fibroblast signature and in genes relatively up-regulated by activin in wound fibroblasts (Fig. 3a). Leading edge analysis identified commonly enriched ECM-related genes, including those encoding lysyl oxidase (*Lox*), procollagen-lysine,2-oxoglutarate 5-dioxygenase 2 (*Plod2*), and *Postn* (Supplementary Fig. 3d right). These results suggest that activin co-expressed genes are enriched in wound fibroblasts, which are exposed to increased levels of endogenous activin compared to cells in NS[33], and further up-regulated in these cells by the transgene-derived activin.

Comparison of the activin-regulated genes to gene sets derived from whole wounds sampled at the three main phases of healing[34] identified a strong positive enrichment toward the remodeling phase in Act mice (Fig. 3b), suggesting an early-to-late healing phase polarization of wound fibroblasts by activin. This is supported by the positive enrichment for early- and late-phase myofibroblasts[10] in activin-regulated genes, compared to singular enrichment for the early-phase myofibroblasts in the total wound fibroblast signature (Fig. 3c). Activin-regulated genes specifically enriched for genes characteristic of a fibrosis-associated adipocyte precursor myofibroblast sub-type[13], and for genes overexpressed in cultured human keloid fibroblasts[35,36], in stiff matrix-forming pre-osteoblasts[37], and in scar-forming (engrailed+) fibroblasts[9] (Fig. 3c). Leading edge analyses implicated various co-enriched activin-regulated genes in pathological scar formation, including *Postn*, *Cthrc1* (encoding collagen triple helix repeat containing 1, which is overexpressed in human keloids and keloid-derived fibroblasts[38]) and *Sfrp2* (encoding secreted frizzled-related protein 2, a marker of pro-fibrotic fibroblasts in human skin[39]) in keloid fibroblasts and wound myofibroblasts, and *Lox* and *Plod2* in wound myofibroblasts (Supplementary Fig. 3e).

A comparison to single-cell RNA sequencing data of murine wound stromal cells, which had identified 12 fibroblast clusters in early-phase large excisional wounds[14], showed enrichment of the general wound fibroblast signature for all fibroblast clusters, which highly express *Pdgfra* and localize throughout the wound granulation tissue (Fig. 3d). Activin overexpression enriched for

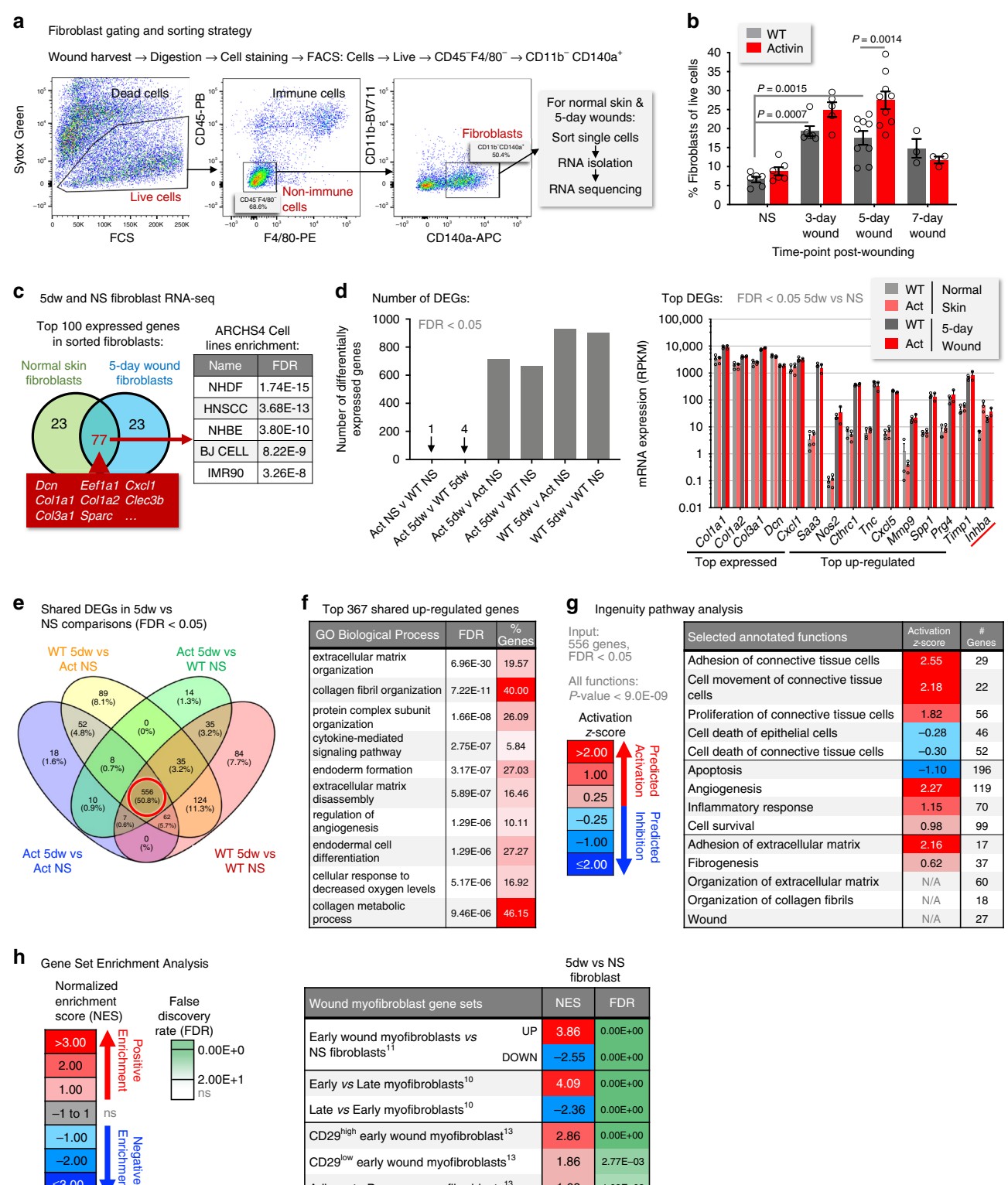

clusters 2 and 8 that localized throughout or in the upper granulation tissue, respectively[14], and which showed enrichment of the fibrosis-associated genes *Aspn*, *Postn*, *Col3a1*, *Lox*, *Cthrc1*, and *Bgn* (encoding biglycan) in cluster 2 and *Col1a1* in cluster 8, among others (Supplementary Fig. 3f).

IPA of activin-regulated genes predicted activation of connective tissue cell movement and adhesion, fibroblast adhesion and proliferation, mineralization, growth of connective tissue, tensile strength of skin, and fibrosis (Fig. 3e, f). Enrichment

analysis with the Reactome database[40], which contains annotations of ECM-related pathways, showed enrichment of genes involved in collagen biosynthesis and modification, fibril assembly, and ECM degradation (Fig. 3g).

To extract the genes responsible for enrichment of ECM-related pathways, we filtered the ranked list of activin-regulated genes to only those defined by the Matrisome database[31] with associated annotation of ECM categories (Fig. 3h). The filtered genes encode proteins involved in different steps of collagen

**Fig. 2 Wound fibroblasts exhibit a distinct transcriptional signature. a** Gating strategy for skin fibroblast isolation. **b** Quantification of fibroblast frequency relative to live cells in unwounded (normal) skin (NS) and in wounds of WT and Act mice. $n = 6, 6, 9, 3$ for WT, $n = 6, 7, 9, 3$ for Act, for NS, 3-, 5-, or 7-day wounds, respectively. **c** Left: Venn diagram showing the top 100 most highly expressed (by RPKM) genes in NS and 5-day wound fibroblasts. Right: Cell line enrichment analysis of the 77 overlapping genes via ARCHS4 (by enrichR), showing top five cell lines with enrichment of these genes; fibroblast-like cells are in bold. **d** Analysis of differentially expressed genes (DEGs). Left: Numbers of DEGs (false discovery rate (FDR) < 0.05) for all individual group comparisons. Right: Absolute expression of selected top-expressed (by RPKM) and top up-regulated (in 5dw vs NS comparisons) genes in the four groups. $n = 3$ for WT_NS, WT_5dw, Act_NS, $n = 2$ for Act_5dw. **e** Venn diagram showing the DEGs (FDR < 0.05) in all 5dw vs NS comparisons. Red circle shows the 556 genes shared between all comparisons. **f** 367 shared up-regulated DEGs from **e** ($Log_2FC > 1$) were subjected to functional enrichment analysis using enrichR. Top 10 Gene Ontology (GO) Biological Processes are shown, with FDR and percentage of input out of total pathway genes (% genes). **g** Shared DEGs from **e** were subjected to Ingenuity Pathway Analysis (IPA). Selected annotated functions are shown with Activation $Z$-scores and numbers of input genes enriched in the respective functions (# genes). $Z$-score > 0 (red): predicted activation of function; $Z$-score < 0 (blue): predicted inhibition of function; $P$ value < 9.0E-09 for all shown functions. **h** The 5-day wound fibroblast signature was subjected to GSEA against gene sets from wound myofibroblasts. Normalized Enrichment Scores (NES) and FDR values are shown; NES > 1: positive enrichment (red), NES < −1: negative enrichment (blue); FDR are color coded based on statistical significance (green). Wound myofibroblast gene sets include genes up/down-regulated in α-SMA$^+$ myofibroblasts from 7-day small excisional wounds[11]; genes up-regulated in α-SMA$^+$ myofibroblasts from 12- or 26-day large excisional wounds[10] and CD29$^{high}$, CD29$^{low}$, or adipocyte precursor myofibroblasts from 5-day small excisional wounds[13]. Graphs show mean±SEM and $P$ values; two-way ANOVA and Bonferroni's multiple comparison post hoc tests (**a**). All $n$ numbers indicate biological replicates. Source data are provided as a Source Data file (Fig. 2).

biosynthesis[38,41–45], including procollagen alpha-chains (*Col3a1*, *Col5a2*, *Col5a3*, *Col7a1*), prolyl hydroxylases (*P4ha2*, *P4ha3*), LH2 (*Plod2*), and LOX (*Lox*) (Fig. 3h). In addition, the list contained genes encoding proteins involved in collagen synthesis (complement C1q tumor necrosis factor-related protein 3 (*C1qtnf3*), Cthrc1 (*Cthrc1*) and WNT1-inducible-signaling-pathway protein 1 (*Wisp1*)), as well as proteins involved in collagen binding (asporin, biglycan, proteoglycan 4, and periostin (*Aspn*, *Bgn*, *Prg4*, *Postn*)) (Fig. 3h). To determine the clinical significance of these findings, we queried transcriptomes of human keloids compared to matched non-lesional skin[46] and of mechanically stressed mouse wounds that progress to hypertrophic scars[47], and found that many of the activin-regulated matrix genes were among the top up-regulated genes in these datasets, including *Postn*, *Plod2*, *Wisp1*, and *Lox*, with *Col3a1*, *Aspn*, and *Cthrc1* being also in the annotated disease-gene association network for keloids[48] (Fig. 3i).

Finally, we used the STRING database[49] to create a data-driven gene–gene interaction network, which connected all activin-enriched ECM regulators, glycoproteins and proteoglycans through the collagens as hubs, and we confirmed the strong association of activin-regulated non-collagenous ECM genes in collagen-related pathways (Supplementary Fig. 3g).

**Activin is a direct regulator of ECM genes in fibroblasts.** We next tested if some genes that are up-regulated in Act mice in vivo are directly regulated by activin. Treatment of primary mouse neonatal skin fibroblasts with activin A indeed induced the expression of *Col1a1*, *Wisp1*, *Postn*, and *Aspn* within 1.5 or 3 h (Fig. 4a). We next cultured fibroblasts from 5-day wounds and found that activin treatment of these cells for 3 h promoted expression of *Wisp1* in cells from WT and Act mice. Interestingly, expression of *Col1a1* and *Aspn* was already significantly higher in fibroblasts from Act vs WT wounds, but was not further up-regulated by exogenous activin (Fig. 4b). This finding suggests that activin induces epigenetic alterations in wound fibroblasts, which are preserved upon culturing. Indeed, cultured WT wound fibroblasts were more activated compared to fibroblasts derived from adult unwounded skin, as evidenced by higher baseline expression of *Col1a1* and *Lox* (Supplementary Fig. 4a).

Due to the robust upregulation of *Aspn* and *Postn* expression by activin in cultured fibroblasts, we determined if these genes are direct activin targets via Smad2/3 signaling. Indeed, several conserved Smad-binding elements (SBE) are present in the promoter and first intron regions of these genes (Supplementary

Fig. 4b). Chromatin immunoprecipitation (ChIP) using lysates from activin vs vehicle-treated immortalized fibroblasts identified that activin promoted binding of Smad2/3 to the conserved SBE in the promoter region of *Postn* as well as to the conserved SBE in the promoter region of *Aspn* (Fig. 4c, Supplementary Fig. 4c). The effect of activin on ECM gene expression was verified at the protein level. Thus, a 24-h treatment of primary human fibroblasts promoted deposition of collagen type I and fibronectin (Fig. 4d).

To study activin regulation of ECM deposition in a system more similar to the wound microenvironment, we established HaCaT keratinocytes with doxycycline (Dox)-inducible over-expression of activin (ActOE) or transduced with empty vector (EV) (Supplementary Fig. 5a, b) and used them for co-culture experiments with immortalized mouse fibroblasts with constitutive expression of GFP and Dox-inducible expression of a dominant-negative activin receptor IB (dnActRIB) mutant[28]. After 7 days of co-culture in starvation conditions (1% FBS), ActOE keratinocytes promoted collagen type I and fibronectin deposition by control (EV) fibroblasts (Fig. 5a, Supplementary Fig. 5c). Importantly, the effect of activin was abolished when keratinocytes were co-cultured with dnActRIB-expressing fibroblasts. The activin-negating effect of dnActRIB was even observed in the presence of 10% FBS (Fig. 5b, Supplementary Fig. 5d), a condition more closely resembling the wound environment[50]. Periostin and asporin deposition was also significantly increased in ActOE/EV vs EV/EV co-cultures and decreased in EV/dnActRIB and ActOE/dnActRIB co-cultures, especially compared to ActOE/EV (Fig. 5c, d, Supplementary Fig. 5e, f). There was a particularly robust deposition of collagen type I and asporin by fibroblasts (GFP$^+$ cells) in proximity to keratinocytes of ActOE/EV, but not of ActOE/dnAlk4 co-cultures (Fig. 5e, f). These findings suggest strong pro-fibrotic paracrine effects of keratinocyte-derived activin on nearby fibroblasts.

Upon FACS-sorting of GFP$^+$ fibroblasts and GFP$^-$ keratinocytes from the co-cultures (Supplementary Fig. 5g) we confirmed *INHBA* overexpression in the keratinocytes of ActOE/EV and ActOE/dnActRIB co-cultures (Fig. 5g) and overexpression of *dnActRIB* relative to endogenous *ActRIB* in the fibroblasts of EV/dnActRIB and ActOE/dnActRIB co-cultures (Fig. 5h). Fibroblasts from ActOE/EV showed upregulation of *Aspn*. Importantly, expression of all analyzed genes, including *Lox* and *Postn*, was significantly down-regulated in co-cultures with dnActRIB fibroblasts (Fig. 5h), suggesting that pro-fibrotic ECM gene expression is dependent on activin receptor signaling in fibroblasts.

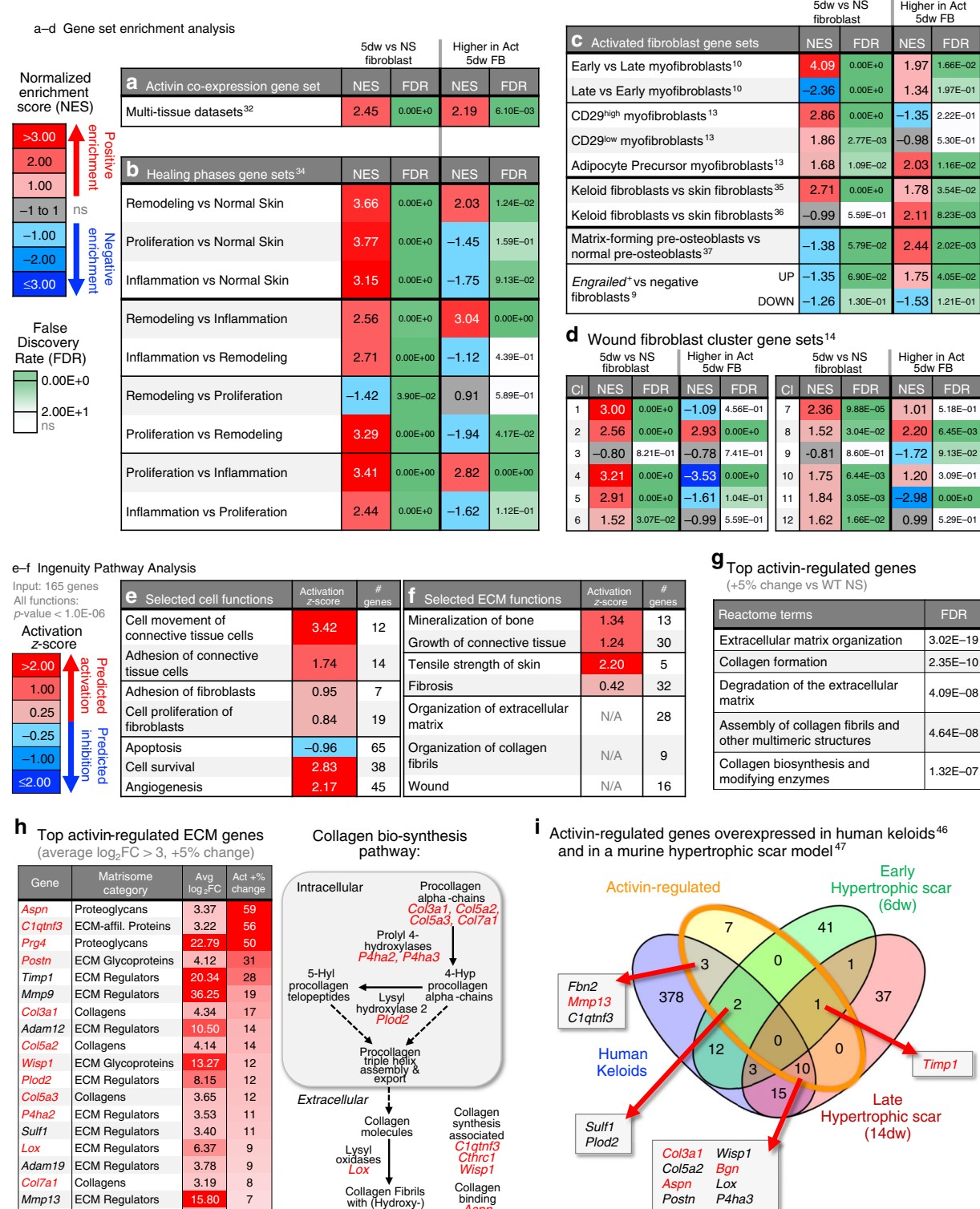

**a–d Gene set enrichment analysis**

**Normalized enrichment score (NES)**

| | |
|---|---|
| >3.00 | Positive enrichment |
| 2.00 | |
| 1.00 | |
| –1 to 1 | ns |
| –1.00 | Negative enrichment |
| –2.00 | |
| ≤3.00 | |

**False Discovery Rate (FDR)**

| | |
|---|---|
| 0.00E+0 | |
| 2.00E+1 | ns |

**a** Activin co-expression gene set

| | 5dw vs NS fibroblast | | Higher in Act 5dw FB | |
|---|---|---|---|---|
| Activin co-expression gene set | NES | FDR | NES | FDR |
| Multi-tissue datasets[32] | 2.45 | 0.00E+0 | 2.19 | 6.10E-03 |

**b** Healing phases gene sets[34]

| Healing phases gene sets[34] | NES | FDR | NES | FDR |
|---|---|---|---|---|
| Remodeling vs Normal Skin | 3.66 | 0.00E+0 | 2.03 | 1.24E-02 |
| Proliferation vs Normal Skin | 3.77 | 0.00E+0 | –1.45 | 1.59E-01 |
| Inflammation vs Normal Skin | 3.15 | 0.00E+0 | –1.75 | 9.13E-02 |
| Remodeling vs Inflammation | 2.56 | 0.00E+0 | 3.04 | 0.00E+00 |
| Inflammation vs Remodeling | 2.71 | 0.00E+00 | –1.12 | 4.39E-01 |
| Remodeling vs Proliferation | –1.42 | 3.90E-02 | 0.91 | 5.89E-01 |
| Proliferation vs Remodeling | 3.29 | 0.00E+00 | –1.94 | 4.17E-02 |
| Proliferation vs Inflammation | 3.41 | 0.00E+00 | 2.82 | 0.00E+00 |
| Inflammation vs Proliferation | 2.44 | 0.00E+0 | –1.62 | 1.12E-01 |

**c** Activated fibroblast gene sets

| | 5dw vs NS fibroblast | | Higher in Act 5dw FB | |
|---|---|---|---|---|
| Activated fibroblast gene sets | NES | FDR | NES | FDR |
| Early vs Late myofibroblasts[10] | 4.09 | 0.00E+0 | 1.97 | 1.66E-02 |
| Late vs Early myofibroblasts[10] | –2.36 | 0.00E+0 | 1.34 | 1.97E-01 |
| CD29high myofibroblasts[13] | 2.86 | 0.00E+0 | –1.35 | 2.22E-01 |
| CD29low myofibroblasts[13] | 1.86 | 2.77E-03 | –0.98 | 5.30E-01 |
| Adipocyte Precursor myofibroblasts[13] | 1.68 | 1.09E-02 | 2.03 | 1.16E-02 |
| Keloid fibroblasts vs skin fibroblasts[35] | 2.71 | 0.00E+0 | 1.78 | 3.54E-02 |
| Keloid fibroblasts vs skin fibroblasts[36] | –0.99 | 5.59E-01 | 2.11 | 8.23E-03 |
| Matrix-forming pre-osteoblasts vs normal pre-osteoblasts[37] | –1.38 | 5.79E-02 | 2.44 | 2.02E-03 |
| Engrailed+ vs negative fibroblasts[9] UP | –1.35 | 6.90E-02 | 1.75 | 4.05E-02 |
| Engrailed+ vs negative fibroblasts[9] DOWN | –1.26 | 1.30E-01 | –1.53 | 1.21E-01 |

**d** Wound fibroblast cluster gene sets[14]

| Cl | 5dw vs NS fibroblast NES | FDR | Higher in Act 5dw FB NES | FDR | Cl | 5dw vs NS fibroblast NES | FDR | Higher in Act 5dw FB NES | FDR |
|---|---|---|---|---|---|---|---|---|---|
| 1 | 3.00 | 0.00E+0 | –1.09 | 4.56E-01 | 7 | 2.36 | 9.88E-05 | 1.01 | 5.18E-01 |
| 2 | 2.56 | 0.00E+0 | 2.93 | 0.00E+0 | 8 | 1.52 | 3.04E-02 | 2.20 | 6.45E-03 |
| 3 | –0.80 | 8.21E-01 | –0.78 | 7.41E-01 | 9 | -0.81 | 8.60E-01 | –1.72 | 9.13E-02 |
| 4 | 3.21 | 0.00E+0 | –3.53 | 0.00E+0 | 10 | 1.75 | 6.44E-03 | 1.20 | 3.09E-01 |
| 5 | 2.91 | 0.00E+0 | –1.61 | 1.04E+0 | 11 | 1.84 | 3.05E-03 | –2.98 | 0.00E+0 |
| 6 | 1.52 | 3.07E-02 | –0.99 | 5.59E-01 | 12 | 1.62 | 1.66E-02 | 0.99 | 5.29E-01 |

**e–f Ingenuity Pathway Analysis**

Input: 165 genes
All functions: p-value < 1.0E-06

**Activation z-score**

| | |
|---|---|
| >2.00 | Predicted activation |
| 1.00 | |
| 0.25 | |
| –0.25 | Predicted inhibition |
| –1.00 | |
| ≤2.00 | |

**e** Selected cell functions

| Selected cell functions | Activation z-score | # genes |
|---|---|---|
| Cell movement of connective tissue cells | 3.42 | 12 |
| Adhesion of connective tissue cells | 1.74 | 14 |
| Adhesion of fibroblasts | 0.95 | 7 |
| Cell proliferation of fibroblasts | 0.84 | 19 |
| Apoptosis | –0.96 | 65 |
| Cell survival | 2.83 | 38 |
| Angiogenesis | 2.17 | 45 |

**f** Selected ECM functions

| Selected ECM functions | Activation z-score | # genes |
|---|---|---|
| Mineralization of bone | 1.34 | 13 |
| Growth of connective tissue | 1.24 | 30 |
| Tensile strength of skin | 2.20 | 5 |
| Fibrosis | 0.42 | 32 |
| Organization of extracellular matrix | N/A | 28 |
| Organization of collagen fibrils | N/A | 9 |
| Wound | N/A | 16 |

**g** Top activin-regulated genes
(+5% change vs WT NS)

| Reactome terms | FDR |
|---|---|
| Extracellular matrix organization | 3.02E−19 |
| Collagen formation | 2.35E−10 |
| Degradation of the extracellular matrix | 4.09E−08 |
| Assembly of collagen fibrils and other multimeric structures | 4.64E−08 |
| Collagen biosynthesis and modifying enzymes | 1.32E−07 |

**h** Top activin-regulated ECM genes
(average log2FC > 3, +5% change)

| Gene | Matrisome category | Avg log2FC | Act +% change |
|---|---|---|---|
| Aspn | Proteoglycans | 3.37 | 59 |
| C1qtnf3 | ECM-affil. Proteins | 3.22 | 56 |
| Prg4 | Proteoglycans | 22.79 | 50 |
| Postn | ECM Glycoproteins | 4.12 | 31 |
| Timp1 | ECM Regulators | 20.34 | 28 |
| Mmp9 | ECM Regulators | 36.25 | 19 |
| Col3a1 | Collagens | 4.34 | 17 |
| Adam12 | ECM Regulators | 10.50 | 14 |
| Col5a2 | Collagens | 4.14 | 14 |
| Wisp1 | ECM Glycoproteins | 13.27 | 12 |
| Plod2 | ECM Regulators | 8.15 | 12 |
| Col5a3 | Collagens | 3.65 | 12 |
| P4ha2 | ECM Regulators | 3.53 | 11 |
| Sulf1 | ECM Regulators | 3.40 | 11 |
| Lox | ECM Regulators | 6.37 | 9 |
| Adam19 | ECM Regulators | 3.78 | 9 |
| Col7a1 | Collagens | 3.19 | 8 |
| Mmp13 | ECM Regulators | 15.80 | 7 |
| Bgn | Proteoglycans | 4.51 | 7 |
| Fbn2 | ECM Glycoproteins | 5.57 | 7 |
| P4ha3 | ECM Regulators | 10.61 | 7 |
| Cthrc1 | ECM Glycoproteins | 82.97 | 5 |
| Ltbp2 | ECM Glycoproteins | 5.87 | 5 |

**Collagen bio-synthesis pathway:**

Intracellular
Procollagen alpha -chains
*Col3a1, Col5a2, Col5a3, Col7a1*
Prolyl 4-hydroxylases *P4ha2, P4ha3*
4-Hyp procollagen alpha -chains
5-Hyl procollagen telopeptides
Lysyl hydroxylase 2 *Plod2*
Procollagen triple helix assembly & export

Extracellular
Collagen molecules
Lysyl oxidases *Lox*
Collagen Fibrils with (Hydroxy-) allysines
Collagen cross-linking pathways

Collagen synthesis associated *C1qtnf3 Cthrc1 Wisp1*
Collagen binding *Aspn Bgn Prg4 Postn*

**i** Activin-regulated genes overexpressed in human keloids[46] and in a murine hypertrophic scar model[47]

Activin-regulated
Early Hypertrophic scar (6dw)
Human Keloids
Late Hypertrophic scar (14dw)

*Fbn2 Mmp13 C1qtnf3*
*Sulf1 Plod2*
*Timp1*
*Col3a1 Wisp1 / Col5a2 Bgn / Aspn Lox / Postn P4ha3 / Cthrc1 Adam12*

7, 3, 0, 41, 1, 378, 2, 37, 12, 1, 3, 10, 0, 15, 0

*Red*: enriched in disease (keloid)-gene association network[48]

**Activin stimulates glycoprotein and proteoglycan production.** The transcriptomic and in vitro data strongly suggested that the histological differences between Act and WT wounds are at least in part associated with increased deposition of fibrosis-associated glycoproteins and proteoglycans. Indeed, Alcian Blue glycan staining of 5-day wounds showed larger areas positive for carboxylated/sulfated acidic glycans in wounds of Act mice (Fig. 6a), but no discernible differences in sulfated acidic glycans (Supplementary Fig. 6a).

Immunostaining showed increased abundance of fibronectin and asporin in the centers of 5d and 10d wounds of Act vs WT mice (Fig. 6b, c). Periostin accumulated at the wound margins in

**Fig. 3 Activin induces a pro-fibrotic gene expression signature in fibroblasts. a–d** Genes regulated in 5dw vs NS fibroblasts and pre-ranked list of genes relatively up-regulated in Act 5dw/WT NS vs WT 5dw/WT NS (see Supplementary Fig. 3a, b) were subjected to GSEA against custom gene sets. NES and FDR values are shown. FDR are color-coded based on statistical significance (green). **a** Activin co-expression gene set derived from the SEEK database of multi-tissue gene expression datasets[32]. **b** Healing phase gene sets derived from microarray data of small excisional mouse wounds at different healing time points[34]. **c** Activated fibroblast gene sets include genes up-regulated in: (i) α-SMA-positive myofibroblasts from 12- or 26-day large excisional wounds[10]; (ii) CD29[high], CD29[low], or adipocyte precursor myofibroblasts from 5-day small excisional wounds[13]; (iii) human keloid vs normal skin fibroblasts[35,36]; (iv) matrix-forming vs normal pre-osteoblasts[37]; and genes up- or down-regulated in scar-forming engrailed[+] vs engrailed[−] fibroblasts[9]. **d** Twelve wound fibroblast cluster gene sets identified by single-cell RNA sequencing of fibroblasts from 12-day large excisional wounds[14]. **e, f** Genes relatively up-regulated in 5dw of Act mice were subjected to IPA. Selected annotated cell functions (**e**) and ECM functions (**f**) were extracted and are shown with respective activation Z-scores and numbers of input genes enriched in the respective functions (# genes). $P$ value < 1.0E-06 for all shown functions. **g** Top activin-regulated genes (at least 5% change vs WT NS) were subjected to functional enrichment analysis using enrichR, and the top five Reactome terms are shown with respective FDR values. **h** Top activin-regulated genes (more than 5% change vs WT) were filtered by the Matrisome database. Left: Top up-regulated (in 5dw vs NS, $\log_2$FC > 3, FDR < 0.05) ECM genes are shown with respective Matrisome category, average $\log_2$FC (5dw vs NS), and the relative increase in upregulation in Act vs WT 5dw fibroblasts (Act +% change). Numerical values are color coded based on magnitude (red background). Right: Genes in red are associated with the collagen biosynthesis pathway. **i** Venn diagram comparison of activin-regulated ECM genes (orange oval) and genes significantly overexpressed in keloids[46] (blue oval) and in a murine hypertrophic scar model (6-day wound, green oval; 14-day wound, red oval[47]), with additional enrichment in the keloid gene association network[48], red text). Source data are provided as a Source Data file (Fig. 3).

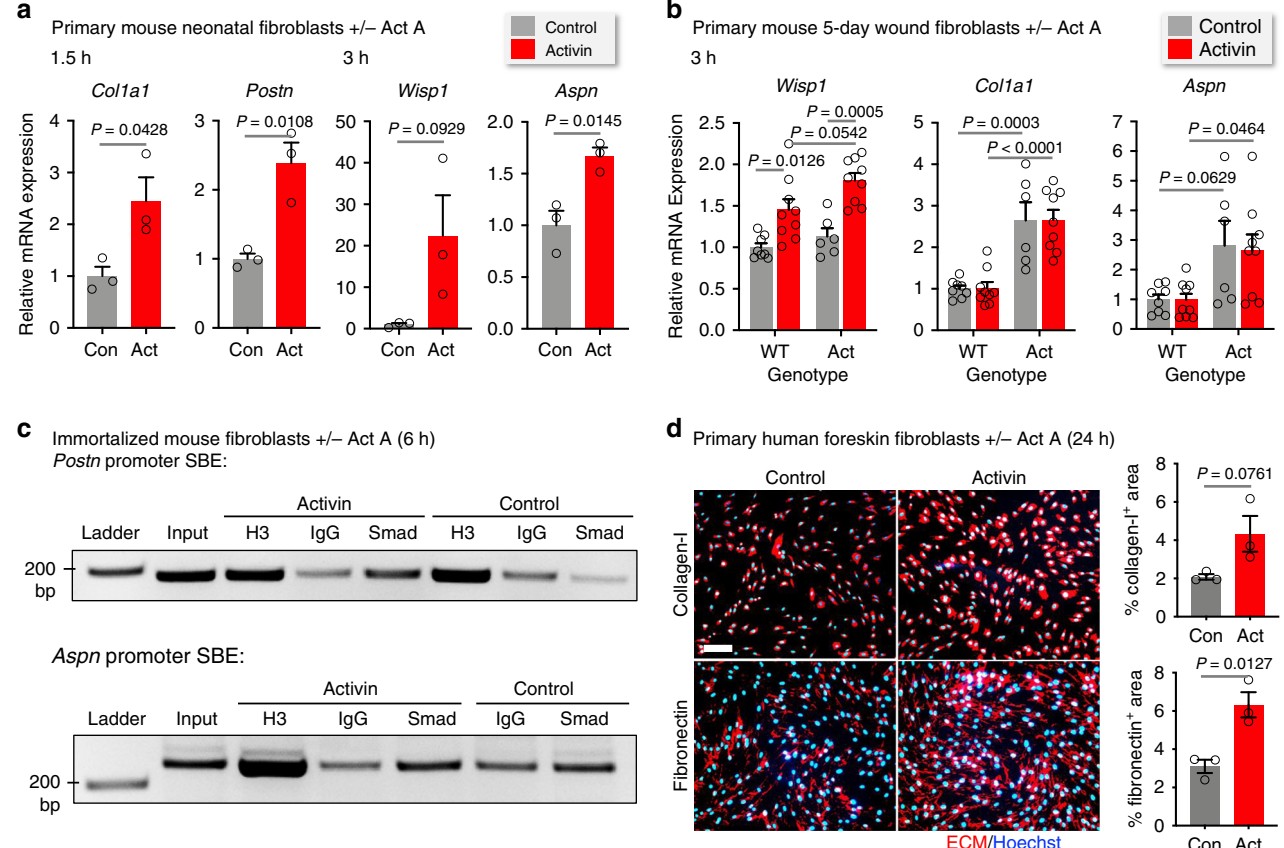

**Fig. 4 Activin directly regulates pro-fibrotic ECM gene expression in fibroblasts. a** qRT-PCR (relative to *Rps29*) using RNA from primary murine neonatal fibroblasts treated with activin A (20 ng/ml) or vehicle for 1.5 and 3 h. $n = 3$. **b** qRT-PCR using RNA from murine fibroblasts from 5-day wounds of adult WT and Act mice treated with activin A (20 ng/ml) or vehicle for 3 h. $n = 8, 9, 6, 9$ for WT-control, WT-Activin, Act-Control, and Act-Activin, respectively. **c** Chromatin immunoprecipitation from lysates of immortalized mouse fibroblasts treated with activin A (20 ng/ml) or vehicle for 6 h. Antibodies against anti-histone H3 (H3) or Smad2/3 (Smad) or normal rabbit IgG (IgG) were used. The bound DNA was amplified using primers spanning conserved Smad-binding elements (SBEs) in the promoter regions of *Postn* and *Aspn* (see Supplementary Fig. 4b). Representative agarose gels are shown, with the size marker and respective inputs. The experiment was repeated twice with similar results (see also Supplementary Fig. 4c). **d** Representative photomicrographs of human primary foreskin fibroblasts treated with activin A (20 ng/ml) or vehicle for 24 h and stained for collagen type I or fibronectin (red) (nuclei were counterstained with Hoechst (blue)). Bar graphs show quantification of the percentage of the stained area. Scale bars: 200 μm. $n = 3$. Graphs show mean ± SEM and $P$ values; mean expression levels in control cells (**a**) or control cells in WT mice (**b**) were set to 1; two-tailed Student's *t*-test (**a, d**), one-way ANOVA and Tukey's multiple comparison post hoc tests (**b**). All $n$ numbers indicate biological replicates. Source data are provided as a Source Data file (Fig. 4).

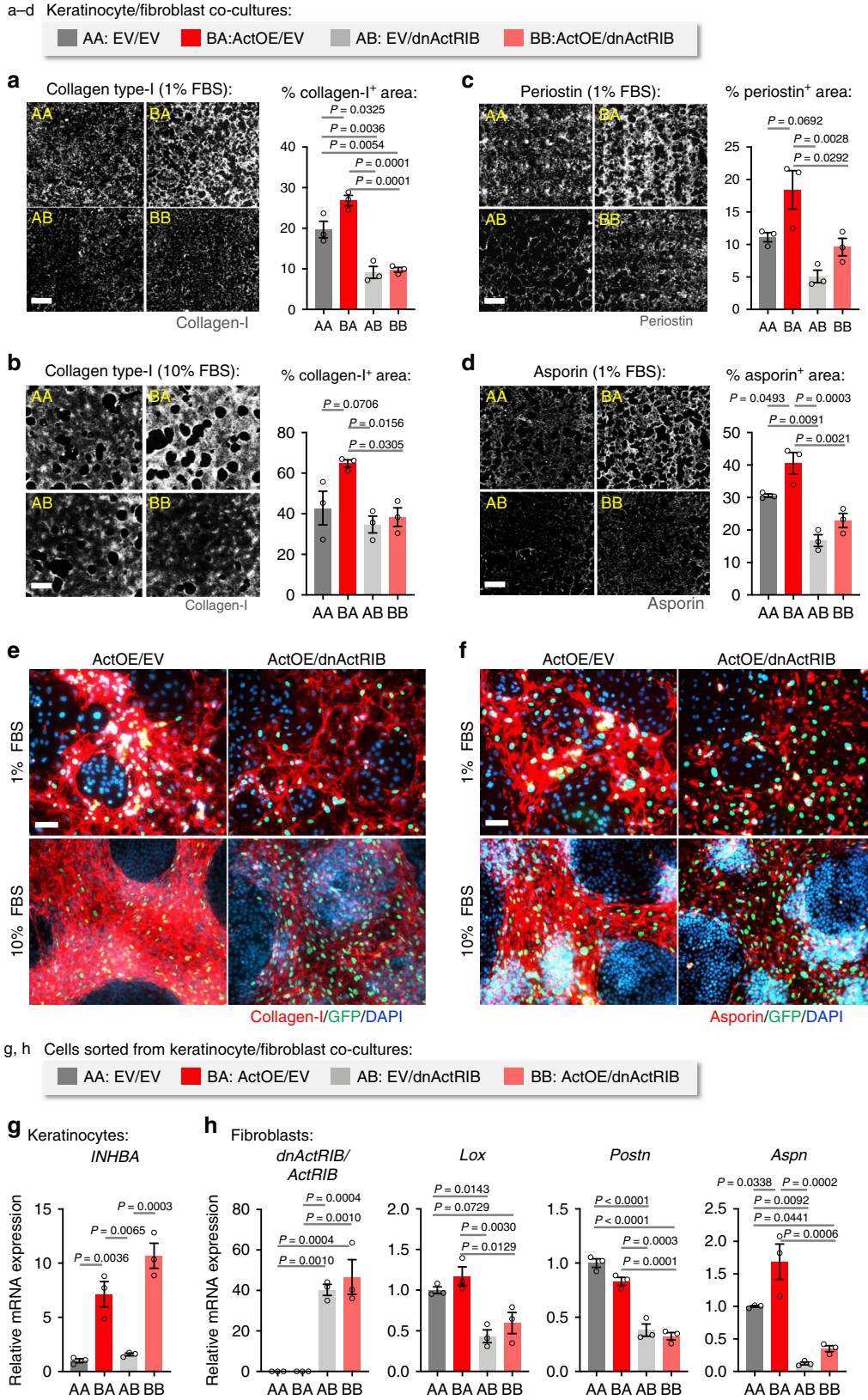

**a–d** Keratinocyte/fibroblast co-cultures:

**g, h** Cells sorted from keratinocyte/fibroblast co-cultures:

5d wounds and, together with asporin, throughout the late granulation tissue of 10d wounds, and both showed higher abundance in Act mice (Fig. 6d, e). In 5d wounds, asporin and periostin deposition was most pronounced in proximity to the activin-overexpressing keratinocytes of the hyperproliferative epithelium (Supplementary Fig. 6b, c).

**Activin triggers altered collagen maturation pathways**. To quantify the alterations in the matrisome, we performed biochemical analyses of collagen and non-collagenous proteins on whole wound samples. Total collagen increased gradually from unwounded skin to the late remodeling phase, and it was significantly higher at 42d post-injury in Act mice (Fig. 7a). In

**Fig. 5 Activin regulates pro-fibrotic ECM deposition by fibroblasts. a–d** HaCaT keratinocytes transduced with lentiviruses allowing Dox-inducible overexpression of *INHBA* (ActOE) or empty vector (EV) were co-cultured with murine immortalized fibroblasts (GFP-expressing) transduced with lentiviruses allowing Dox-inducible expression of dnActRIB or with EV-transduced fibroblasts for 7 days in 1% or 10% FBS. Co-cultures were stained for collagen type I (**a**, **b**), periostin, and asporin (**c**, **d**). The experiment was repeated twice with similar results. Representative photomicrographs of ECM staining for each group. Bar graphs show quantification of the percentage of the stained area on whole co-culture coverslips. Scale bars: 1000 μm. *n* = 3. **e**, **f** Representative high magnification photomicrographs of collagen type I (**e**) and asporin (**f**) staining of ActOE/EV and ActOE/dnActRIB co-cultures in the presence of 1% or 10% FBS. GFP-positive fibroblasts are shown (green); nuclei were counterstained with DAPI (blue). Scale bars: 100 μm. **g**, **h** GFP-expressing fibroblasts were sorted from GFP-negative HaCaT keratinocytes after 8 days co-culture, and gene expression was analyzed by qRT-PCR relative to *RPL27* for keratinocytes (**g**) or *Rps29* for fibroblasts (**h**). *n* = 3. The experiment was repeated twice with similar results. Graphs show mean ± SEM and *P* values; mean expression levels in AA cultures were set to 1 in **g** and **h**; one-way ANOVA and Tukey's multiple comparison post hoc tests (**a–d**, **g**, **h**). All *n* numbers indicate biological replicates. AA, EV/EV; BA, ActOE/EV; AB, EV/dnActRIB; BB, ActOE/dnActRIB. Source data are provided as a Source Data file (Fig. 5).

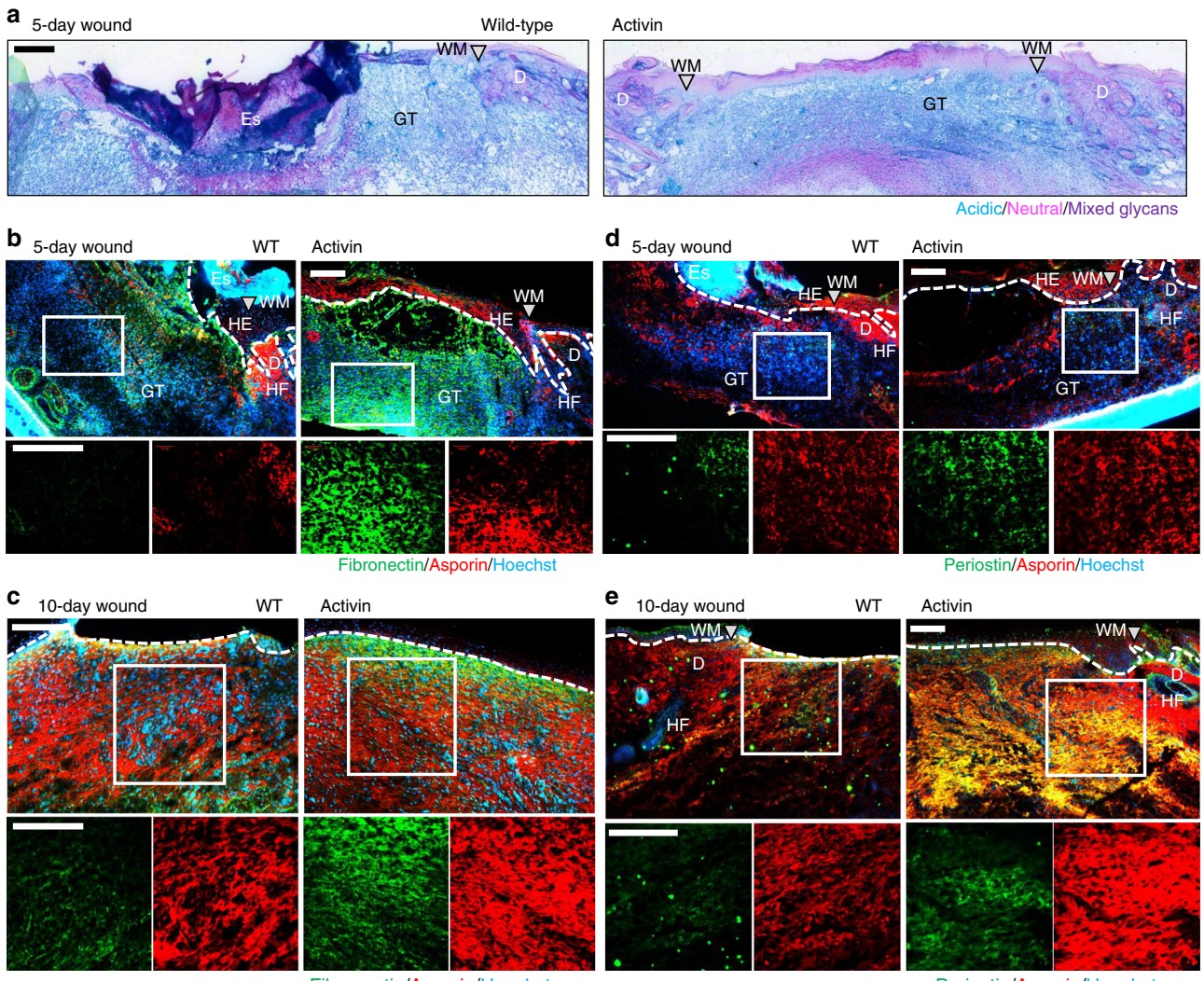

**Fig. 6 Activin promotes deposition of wound glycoproteins and proteoglycans. a** Representative photomicrographs of Alcian Blue/PAS-stained 5-day wounds show larger areas of acidic (blue) glycans in the wound center of Act vs WT mice. **b–e** Representative photomicrographs of sections from 5-day (**b**, **d**) and 10-day wounds (**c**, **e**) of WT and Act mice stained with antibodies against fibronectin (green) and asporin (red) (**b**, **c**) or periostin (green) and asporin (red) (**d**, **e**). Nuclei were counterstained with Hoechst (blue). Insets indicate the areas within the wound bed centers that are shown at higher magnification below. Stainings were performed on *n* = 5 wounds from five mice for 5-day wounds and *n* = 3 for 10-day wounds with similar results. Gray triangles represent wound margins (WM); HE hyperproliferative epithelium, HF hair follicle, D dermis, Es eschar, GT granulation tissue. White dotted lines indicate the border between the dermis/granulation tissue and the epidermis or wound epithelium. Scale bars: 200 μm.

contrast, the bulk amount of non-collagenous proteins was increased in the early phase and strikingly larger in Act wounds, both at early time points (3–5d) and in unwounded skin (Fig. 7b). This result points to large contributions of glycoproteins and proteoglycans in Act mice to the overall proteome of early wounds, thus confirming our histological findings. Calculation of the ratio of collagen per total protein normalized to NS showed a marked increase in relative collagen content in Act mice at all

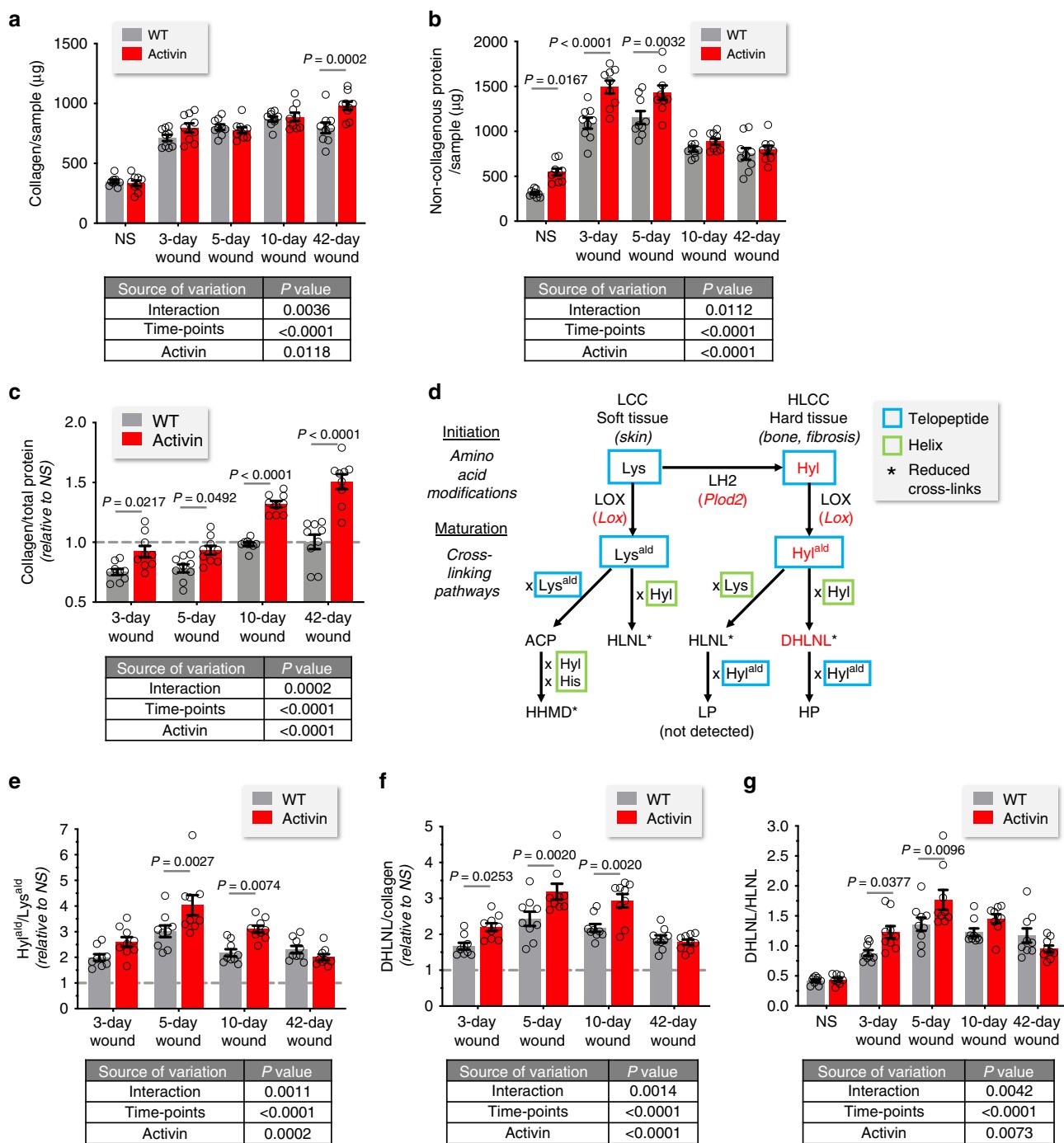

**Fig. 7 Activin promotes collagen maturation in healing wounds.** Biochemical analysis of collagen and non-collagenous proteins using whole lysate from unwounded skin (NS) and from wounds at day 3, 5, 10, and 42 post-injury. **a** Amount of total collagen per sample. **b** Amount of total non-collagenous protein per sample. **c** Amount of collagen per total protein relative to NS (gray dashed line). **d** Diagram outlining the collagen cross-linking initiation and maturation pathways (adapted from ref. [79]), the enzymes that are responsible for the number (LOX) and the pattern (LH2) of the cross-links, and the cross-links quantified in our analysis. Modifications that are increased in wounds of Act mice are in red. **e** Hyl$^{ald}$/Lys$^{ald}$ ratio relative to NS (gray dashed line). **f** DHLNL/collagen ratio relative to NS (gray dashed line). **g** DHLNL/HLNL cross-link ratio. Graphs show mean ± SEM and P values; $n = 9$ biological replicates for mice of both genotypes at all time points; two-way ANOVA and Bonferroni's multiple comparison post hoc tests; results of two-way ANOVA are shown below each graph (first factor: time point, second factor: activin overexpression). HLCC hydroxylysine aldehyde derived collagen cross-links, LCC lysine aldehyde-derived collagen cross-links, Lys lysine, Hyl hydroxylysine, LH lysyl hydroxylase, LOX lysyl oxidase, Lys$^{ald}$ lysine aldehyde, Hyl$^{ald}$ hydroxylysine aldehyde, ACP aldol condensation product, HLNL hydroxylysinonorleucine, DHLNL dihydroxylysinonorleucine, HHMD histidinohydroxymerodesmosine, LP lysylpyridinoline, HP hydroxylysylpyridinoline. Source data are provided as a Source Data file (Fig. 7).

time points (Fig. 7c), indicating higher stimulation of collagen synthesis in Act vs WT mice compared to non-collagenous proteins, especially in late wounds. The hydroxylysine (Hyl)/ hydroxyproline (Hyp) ratio, a measure of the degree of lysyl hydroxylation, progressively increased up to 5d, especially in Act mice (Supplementary Fig. 7a).

Biochemical quantification of components of the two main collagen cross-linking pathways[51] (Fig. 7d) demonstrated a progressive increase in the relative hydroxylysine aldehyde (Hyl$^{ald}$)/lysine aldehyde (Lys$^{ald}$) ratio (normalized to NS), a measure of lysyl hydroxylation of collagen molecule telopeptides involved in cross-links by LH2 and of oxidation of telopeptidyl lysine or hyroxylysine residues performed by LOX, up to 5d (Fig. 7e). Consistent with the regulation of Plod2 and Lox by activin, there was a significantly more pronounced peak of Hyl$^{ald}$/Lys$^{ald}$ at 5d in Act vs WT mice, which was sustained until 10d (Fig. 7e). The dihydroxylysinonorleucine (DHLNL)/collagen ratio (normalized to NS), reflecting the generation of a type of cross-link associated with fibrosis and mechanically stiff tissues (Fig. 7d)[51], progressively increased up to 5d in mice of both genotypes. Wounds from Act mice had a more pronounced increase at 3d and a larger peak at 5d, followed by a sustained greater abundance until 10d (Fig. 7f). The hydroxylysinonorleucine (HLNL)/collagen ratio (normalized to NS), reflecting the generation of a collagen cross-link associated with soft tissues such as skin (Fig. 7d)[51], slightly decreased during wound healing, and this was attenuated in Act wounds during the remodeling phase (10d and 42d) (Supplementary Fig. 7b). The DHLNL/HLNL ratio, which quantifies the differential collagen cross-linking pathways in skin and positively correlates with fibrosis and increased tissue stiffness[51–53], was generally elevated in wounds vs NS and significantly higher in 3d and 5d wounds of Act mice (Fig. 7g). By contrast, there were minimal differences in the histodinohydroxymerodesmosine (HHMD) and hydroxyly-sylpyridinoline (HP) cross-links across time points and between genotypes (Supplementary Fig. 7c, d). These biochemical results demonstrate that activin stimulates the fibrotic pathway of collagen cross-linking during wound healing.

**Activin changes the biomechanical properties of skin wounds.** To determine the biomechanical properties of skin wounds and relate them to histological healing parameters and the fibroblast transcriptome, we developed a non-invasive approach based on observation of the wounded region in vivo using a setup comprised of an upright camera equipped with a telecentric lens (Supplementary Fig. 8a). This method allows simultaneous monitoring of the evolution of (i) the wound morphological area; (ii) the skin deformability in/around the wound; and (iii) the extension of the region biomechanically affected by the wound.

Quantification of the visible wound areas confirmed the expected rate of wound closure in mice of both genotypes (Fig. 8a, Supplementary Fig. 8b, c)[54]. Wounds from Act mice were slightly larger at 3d and 10–21d post-injury. Measurement of the visible wound length along the cranial–caudal direction (Supplementary Fig. 8d) confirmed the exacerbated scar formation of Act mice on 10–21d (Fig. 8b). The rates of change of pooled and individually tracked wound lengths were highest at 3–5d, but Act wounds closed at nearly double the rate of WT wounds (Fig. 8c).

We then subjected a tissue region that included the wound to an elongation of about 10%, which is representative of physiological activities and does not affect the healing process, and we monitored the displacement of fiducial points to reconstruct the deformability of the skin (Supplementary Fig. 8a). Analyzing a narrow strip, which was aligned with the

cranial–caudal direction and included both wounded and unwounded skin, we identified a tissue region where the fiducial points underwent reduced displacements in comparison to the unwounded surroundings and defined this as the wound functional length (Supplementary Fig. 8e). Mice of both genotypes showed a clear reduction of this length upon healing, with a strong evolution between 5d and 10d post-injury (Fig. 8d). However, Act mice displayed significantly higher functional length at all time points. Interestingly, this length was only modestly reduced between 3d and 5d in mice of both genotypes (Fig. 8d), in contrast to the clear reduction in the visible wound length (Fig. 8b). Calculation of the ratio of functional to visible wound length indicated a region of biomechanically affected tissue that is 1.5–2.25 times longer than the visible wounds between 3d and 7d (Fig. 8e). This ratio was still 1.3–1.7 up to 21d post-injury in Act mice, while WT mice had no such extension of affected tissue by day 21 (Fig. 8e).

We further reconstructed the movements of fiducial points within a wider region of interest and determined the corresponding heterogeneous state of deformation at 10% overall tissue elongation using a custom-developed algorithm (Supplementary Fig. 8f). This analysis provided visual evidence of a reduced deformability of the healing tissue in mice of both genotypes and at every considered time point as compared to the non-wounded surroundings (Fig. 8f). For comparison purposes, we quantified the deformability of a region of interest selected within the morphologically visible wound (Supplementary Fig. 8f, wound area) throughout healing. Wounds/scars were consistently stiffer than the surrounding non-wounded skin (Fig. 8g), and wounds from WT mice were at least 80% more deformable than those from Act mice at 3–5d post-injury (Fig. 8h). Remarkably, the difference between WT and Act wound/scar deformability increased between d10 and d21 as Act mice developed stiffer scars. Recovery of tissue deformability was slower in wounds of Act mice at 3–5d and 14–21d, but faster at 7–10d post-injury (Fig. 8i).

A comparison of the deformability of wounds across the healing time-course identified significant differences, particularly between days 3–5 and 10–21 (Supplementary Table 1). Systematic analyses (see Methods) confirmed a high degree of measurement reproducibility and robustness with respect to changes in applied global strain and loading rate, which may arise due to manual control of skin elongation (Supplementary Fig. 9a–h). Thus, the proposed method may be used for quantification of longitudinal evolutions of the deformability of wounded skin towards the formation of a scar.

To further analyze the scar tissue, we excised skin specimens at 21d post-injury and performed uniaxial tensile tests in a custom-built setup allowing for local deformation analysis[17]. We quantified the level of deformation in the scar tissue corresponding to a 10% deformation in the nearby-unwounded skin, and confirmed the reduced deformability of Act scars (Fig. 8j). However, comparing the corresponding ultimate loads (i.e. breaking strength) did not significantly differ between genotypes (Fig. 8k), despite the clear reduction in comparison to unwounded WT skin[17]. These results suggest that tissue characterization at such large levels of deformation might not be representative of physiological loading conditions for wounds and scars.

Finally, we performed uniaxial tensile tests on excised unwounded skin samples and observed similar deformability and kinematics behavior (Supplementary Fig. 8i) between mice of both genotypes, despite a thicker skin (~24%) in Act mice (Supplementary Fig. 9j). Therefore, the biomechanical differences between wounds of mice from both genotypes, and thus the stronger scarring phenotype of Act mice, is a characteristic specifically associated with the healing process, in line with our histological, transcriptomic, and biochemical findings.

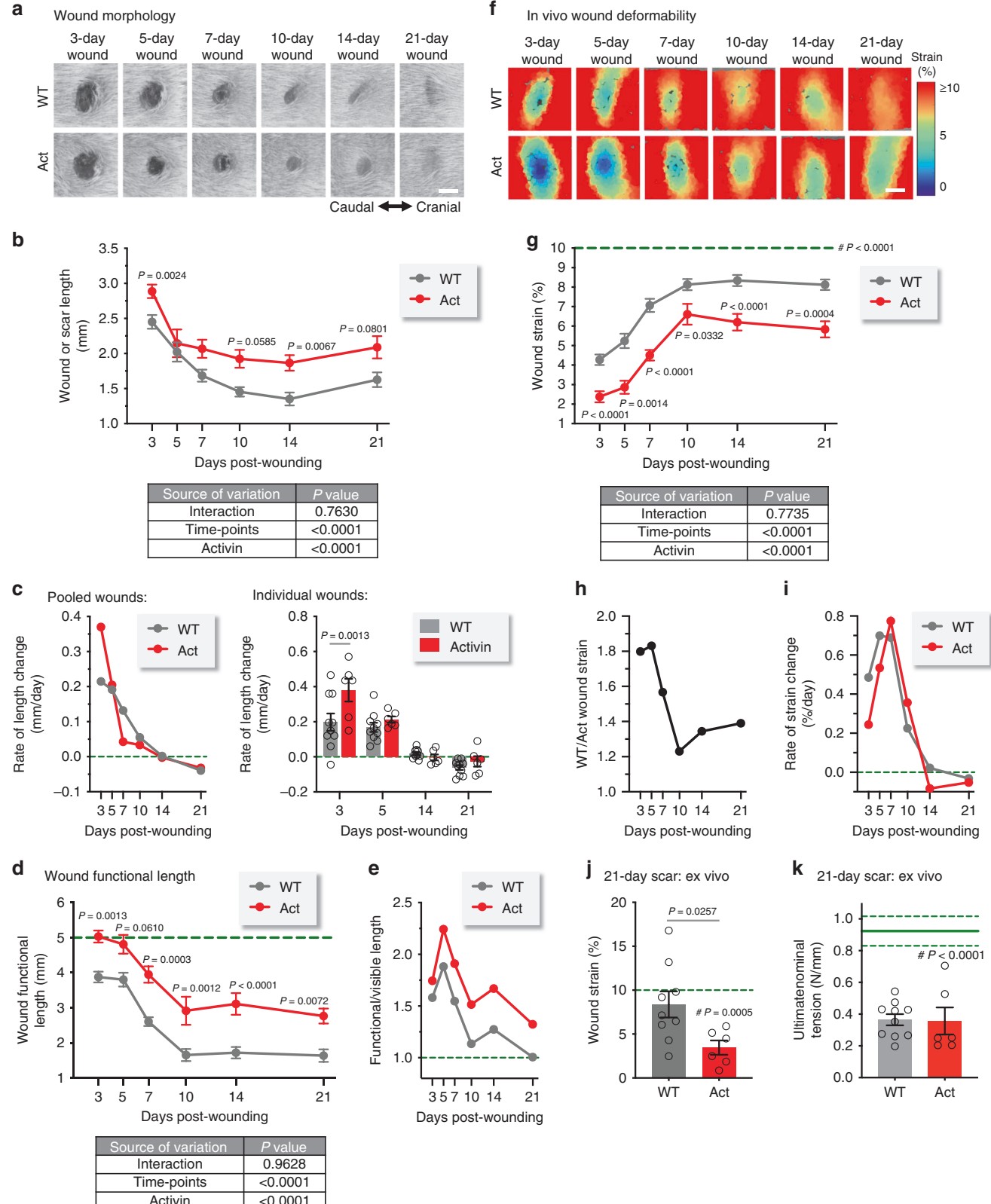

## Discussion

In this study we tracked ECM-related features across scales and time. We show how the early wound fibroblast transcriptome contributes to ECM deposition and dynamics and how these features are affected by activin. As an essential part of this analysis, we developed an in vivo imaging technology that allows determining mechanical features of normal and wounded skin under physiologically relevant conditions. The results provide mechanistic insight into the pro-fibrotic activities of activin and the molecular features that determine the biomechanical properties of skin wounds.

**Fig. 8 Activin affects the biomechanical properties of skin wounds. a** Representative photographs of wounds from WT and Act mice at different days post-injury; scale bar: 2 mm. **b** Quantification of visible wound and scar lengths (see Supplementary Fig. 8b, d). $n = 24, 10, 12, 14, 16, 12$ for WT and $n = 22, 6, 10, 8, 12, 8$ for Act mice in 3-, 5-, 7-, 10-, 14-, or 21-day wounds, respectively. **c** Rates of wound length change; green dashed line: no change. Left: Rates using averaged values of pooled wounds. Right: Rates using individual wounds, which were tracked across multiple consecutive time points; $n = 10$ for WT and $n = 6$ for Act mice for all time points. **d** Quantification of wound functional length (see Supplementary Fig. 8e); green dashed line indicates the 5 mm diameter of the biopsy punch. For $n$ numbers, see **b**. **e** Graph of the ratios of average functional length to corresponding average wound length; green dashed line indicates equal length of functional and visible wounds. **f** Color maps demonstrating in vivo deformability of wounded tissues (see Supplementary Fig. 8f); scale bar: 2 mm. **g** Quantification of in vivo wound strain; green dashed line indicates the behavior of unwounded skin. For $n$ numbers, see **b**. **h** Ratio of average deformation of wounds from WT to Act mice. **i** Rates of localized wound deformation change using averaged values of pooled wounds; green dashed line: no change. **j** Local scar deformability at day 21 post-injury based on ex vivo tensile tests on uniaxial tissue specimens; 10% deformation of the unwounded skin region (green dashed line) is shown for comparison. $n = 9$ for WT and $n = 6$ for Act mice. **k** Tensile strength of WT NS (green line ±SEM) and 21-day scars of WT and Act mice. $n = 8$ for WT NS (from ref. [17]), $n = 10$ for WT and $n = 6$ for Act mice. Graphs show mean ± SEM and $P$ values, with comparisons to NS indicated by #; two-way ANOVA and Bonferroni's multiple comparison post hoc tests (**b, c** right, **d, g**), results of two-way ANOVAs are shown below each graph (first factor: time point, second factor: activin overexpression), cf. Supplementary Table 1 for statistical comparisons across time points; two-tailed Student's $t$-test (**j**); one-sample $t$-test vs NS value (10%) (**g, j**); one-way ANOVA and Tukey's multiple comparison post hoc tests (**k**). All $n$ numbers indicate biological replicates. Source data are provided as a Source Data file (Fig. 8).

During wound healing, fibroblasts gain a transcriptional signature that is enriched for pathways associated with stromal cell proliferation, migration, adhesion to ECM, and ECM formation. Remarkably, a comparison with published wound transcriptomics data suggests the existence of a universal wound-induced fibroblast expression profile that is largely independent of wound size and genetic background. This gene expression signature is most likely determined by growth factors and cytokines released at the wound site[5] as well as by biophysical factors, including mechanical tension[55–57].

Our work shows how changes in the fibroblast transcriptome translate into histological, biochemical, and mechanical alterations in skin wounds. Increased levels of activin promoted the expression of key enzymes involved in collagen biosynthesis and maturation and of ECM glycoproteins and proteoglycans. We identified some of these genes as direct activin targets in fibroblasts, strongly suggesting that their increased levels in the wounds of Act mice are a result of activin-induced fibroblast alterations rather than a consequence of the mild and only transient increase in fibroblast numbers in wounds of Act mice or of the accelerated wound re-epithelialization in these mice. The activin-induced expression profile was enriched in previously identified scar-forming fibroblast sub-types[9,13] and in keloid fibroblasts[35,36]. Since activin is overexpressed in hypertrophic scars, keloids, and in other fibrotic diseases[23,24,29,58], our findings suggest a causative role of increased activin levels in the pathogenesis of a broad spectrum of fibrotic disorders. Therefore, activin and its targets identified in this study qualify as promising biomarkers for diagnosis of skin fibrosis and for therapeutic intervention.

We identified periostin and asporin as direct activin targets in fibroblasts. They enriched in the wound bed of activin-overexpressing mice and were deposited by fibroblasts in an activin receptor-dependent manner. The association of periostin with scar formation is consistent with its overexpression in profibrotic skin fibroblasts[59] and high abundance in hypertrophic scars and keloids[60]. The association of asporin with excessive scarring and its enrichment in datasets of skin fibrosis is consistent with its high abundance in keloid tissue extracts[61] and its overexpression in keloids[62].

There was a clear progression from very stiff to more compliant wound tissue behavior between day 3 and 10 in mice of both genotypes. If collagen was the major determinant of tissue stiffness, the deformability should decrease; therefore, this progression is most likely not a consequence of collagen accumulation. Indeed, we previously demonstrated that collagen present in the wound core at day 7 appears highly crimped and its fibers are

not engaged during skin stretching[17], indicating that it does not significantly contribute to tissue stiffness in the early phase of healing. However, the direct correlation observed between increasing tissue deformability and decreasing non-collagenous protein content points to a contribution of other matrix components, such as glycoproteins and proteoglycans, which were more abundant in the stiffer early wounds from Act mice. During the remodeling phase, the increased stiffness of the resulting scars compared to unwounded skin is consistent with their relatively high collagen content, collagen maturation, and fibrosis-associated cross-linking. The activin-induced alterations of the wound fibroblast transcriptome and matrisome correlated with an increase in fibrosis-associated collagen telopeptide hydroxylation and cross-linking during the mid-to-late phases of healing in the stiffer activin-overexpressing wounds and scars. These features were previously correlated with formation of scars and tissues with strongly compromised mechanical function[53,63,64].

The identification of mechanical properties of wounds was made possible by the development of a non-invasive technology for in vivo optical measurement of skin deformability at physiologically relevant levels of tension, which also allowed us to determine the length of tissue that is mechanically affected by the wound. Importantly, the technology has strong translational potential for the tracking of wound healing and scar formation in humans since: (i) it does not require highly specialized equipment; (ii) the tapes only contact the skin adjacent to the wound; (iii) in-plane displacement requires minimal training, and its analysis is robust toward differences in the amount of applied force or the level of imposed tissue displacement; and (iv) the degree of displacement required for analysis is in a physiologically relevant range and offers direct haptic feedback against possible tissue overloading, thereby minimizing the risk of wound dehiscence.

The measured evolution of wound deformability is not affected by the applied deformation level, as confirmed by comparing the outcomes of data analyses performed at 10% vs 5% global strain. Thus, we could have obtained equivalent results with a much lower stretch, which might constitute an advantage if the method is applied on fragile tissues. However, depending on the image quality, the applied deformation level might affect the reliability of the optical strain analysis.

Current protocols for evaluating wound mechanical properties using in-plane displacement require excision of the wound tissue[17] and/or the use of supra-physiologic levels of strain to measure these properties[15]. Our non-invasive method, though less controlled in terms of applied force, provides for a readout

that takes into account the in vivo properties of wounds relative to their surrounding tissue. This location-based normalization of wound deformation allows for a universal interpretation of localized deformability across time, wound location, and likely between patients. Importantly, in contrast to other non-invasive methods for evaluation of skin stiffness, e.g. methods that measure out-of-plane displacement by indentation[18] or stiffness by suction[65], it does not require physical contact with the fragile wound tissue.

Taken together, this study provides insight into the mechanisms underlying normal and hypertrophic scar formation across scales and identifies important targets for therapeutic intervention. In addition, we present a major technical advance in the biomechanical analysis of wound healing and scar formation with a strong translational potential, which will open avenues for the functional characterization of normal, wounded, and diseased skin.

## Methods

**Animals and wounding experiments**. Act mice, which express *INHBA* under control of the keratin 14 promoter (CD-1 genetic background)[21], were mated with C57BL/6 mice, and their F1 progeny was used for FACS-sorting and RNA sequencing. For all other experiments we used mice in pure CD-1 background. Wild-type littermates were used as controls. Mice were housed under specific pathogen-free (SPF) conditions and received food and water ad libitum. Mouse maintenance and all animal experiments had been approved by the local veterinary authorities (Kantonales Veterinäramt Zürich, Switzerland).

An excisional wound model was used to study wound healing kinetics[54]. Female mice (9–12 weeks old) were anaesthetized by isoflurane inhalation, their backs were shaved and cleaned with 70% ethanol, and full-thickness excisional wounds (5 mm diameter) were generated on either side of the back midline. For biomechanical studies, two wounds were made; for all other studies, four wounds were made. Wounds from each time point were pooled in medium for flow cytometric analysis, or prepared for histology/immunostaining by either fixing in 4% phosphate-buffered paraformaldehyde (PFA) and embedding in paraffin or freezing in tissue freezing medium (Leica Biosystems, Wetzlar, Germany). Wounds were bisected along the cranial–caudal direction and embedded such that sections were always aligned along the same axis and represented the central portion of the wound.

**Isolation and culture of primary dermal fibroblasts**. Primary fibroblasts were isolated from neonatal (P2.5) or adult mouse skin or from 5-day wounds of adult mice. Human primary foreskin fibroblasts were kindly provided by Dr. Hans-Dietmar Beer, University Hospital Zurich. The foreskin had been collected with informed written consent of the parents and upon approval by the local ethics committee. For establishment of primary mouse fibroblasts, dermis was separated from epidermis by incubating with a 5% trypsin/EDTA solution for 1 h at 37 °C and gently peeling away the epidermal layer. The dermis was then minced into small pieces and incubated with collagenase type II solution (500 U/ml; Worthington Biochemical Corporation, Lakewood, NJ) for 1 h at 37 °C with manual agitation every 15 min. The cell suspension was poured through a 100 μm cell strainer and the contents were centrifuged at 1200 r.p.m. for 5 min. The resulting cell pellet was resuspended in Dulbecco modified Eagle's medium (DMEM; Sigma, Munich, Germany), supplemented with 10% FBS/penicillin/streptomycin, and the cells were seeded in culture dishes. Medium was changed the following day and cells were passaged prior to confluency. Absence of mycoplasma was confirmed using the PCR Mycoplasma Test Kit I/C (PromoKine, Heidelberg, Germany).

Cultured primary fibroblasts were starved for 24 h in DMEM/1% FBS before treatment with 20 ng/ml recombinant human activin A (R&D Systems, Minneapolis, MN) in starvation medium for 1.5, 3, or 24 h.

**Cloning, lentiviral production, and transduction of cell lines**. The coding sequences for INHBA or dnActRIB were amplified from plasmids using the primers listed in Supplementary Table 3, cloned into the pENTR/D-TOPO vector (#K240020; Thermo Fisher Scientific, Waltham, MA) and subsequently into the pInducer20 vector (kindly provided by Dr. S. Elledge) using the LR Clonase II Enzyme mix (Thermo Fisher Scientific). HEK293T cells at 30% confluency were transfected overnight in DMEM/10% FBS without P/S using jetPEI Transfection Reagent (Polyplus-Transfection SA, New York, NY). For each 10 cm dish, 5 μg DNA was incubated with 1.75 μg pMD2.G and 3.25 μg pCMV-dR8.91 (Addgene, Cambridge, MA). On the next day, the medium was replaced by DMEM/10% FBS/P/S. Cells were cultured for at least 48 h to allow production of the virus. The supernatant was filtered through a Filtropur S 0.2 μm filter (Sarstedt, Nümbrecht, Germany) and stored at −80 °C.

HaCaT cells (immortalized human keratinocytes) were grown to 60–70% confluency in 6 cm dishes and transduced with pInducer20 lentiviruses[66] that allow Dox-inducible expression of *INHBA* (ActOE) or an empty viral vector (EV) in DMEM/10%FBS/P/S supplemented with 8 μg/ml of polybrene for 6 h at 37°C/5%

$CO_2$. Afterwards, fresh medium with polybrene was added. Two days later, the selection process was started by addition of 1 μg/ml G418/geneticin (Thermo Fisher Scientific). Isolated G418-resistant clones were used for co-culture studies.

Immortalized skin fibroblasts were initially isolated from PDGFRα-eGFP transgenic mice, immortalized via serial passaging, and transduced with pInducer20 lentivirus[66] (as described above) that allow Dox-inducible expression of a dominant-negative activin receptor IB mutant (dnActRIB) or an empty viral vector (EV) (for details and characterization of this cell line, see ref. [28]).

**Co-culture of HaCaT keratinocytes and mouse immortalized fibroblasts**. Four sets of co-cultures were prepared (outlined in Supplementary Fig. 5a): (1) HaCaT EV + immortalized fibroblasts EV (designated AA); (2) HaCaT ActOE+ immortalized fibroblasts EV (designated BA); (3) HaCaT EV+ immortalized fibroblasts dnActRIB (designated AB); (4) HaCaT ActOE + immortalized fibroblasts dnActRIB (designated BB). Doxycycline was added at 2 μg/ml at the beginning of co-culture (or none for controls). For starvation conditions, 1% FBS in DMEM was used; for wound-like conditions, 10% FBS was used. For ECM analyses, 20,000 HaCaT cells were cultured with 20,000 fibroblasts on coverslips in 24-well plates and co-cultures were analyzed after 7 days. For FACS and gene expression analyses, 40,000 HaCaT cells were co-cultured with 40,000 fibroblasts in six-well plates for 8 days, trypsinized, and reconstituted in FACS buffer. GFP+ (fibroblasts) and GFP− (keratinocytes) cells were sorted using the BD FACSAria IIIu equipped with FACSDiva software version 6 (BD Pharmingen, San Diego, CA); sorted cells were re-analyzed to determine >99% purity of sorted cell populations (see Supplementary Fig. 5g).

**Chromatin immunoprecipitation**. SBE were located using FIMO (Find Individual Motif Occurrences; version 5.1.1; http://meme-suite.org/tools/fimo) by inputting the Smad2-binding motif matrix file (Smad2(MAD)/ES-SMAD2-ChIP-Seq (GSE29422)/Homer (Motif 330)) and searching within a 10 kb region upstream of the transcriptional start sites and within the genomic regions of *Postn* and *Aspn* (identified via the UCSC Genome Browser using the *Mus musculus* assembly GRCm38/mm10; genome.ucsc.edu). The identified SBEs were manually verified and the conservation of sequences was evaluated using the UCSC Genome Browser.

Immortalized skin fibroblasts were grown to 80–90% confluency in large culture dishes and starved for 24 h in DMEM/1% FBS before treatment with 20 ng/ml recombinant human activin A (R&D Systems) in starvation medium for 6 h. Approximately $5 \times 10^6$ cells were harvested and DNA/protein complexes crosslinked with 37% PFA for 10 min and quenched in 0.1 M glycine for 10 min. All subsequent buffers were supplemented with 1 mM AEBSF and 1× cOmplete protease inhibitor cocktail (Roche). Cells were washed in ice-cold 1× PBS and nuclei were prepared in TNT buffer (10 mM Tris, 10 mM NaCl, 0.2% TritonX-100, pH 8). Nuclei were resuspended in NUC buffer (10 mM HEPES, 60 mM KCl, 15 mM NaCl, 0.32 mM sucrose, 0.5 mM DTT, pH 7.5) and incubated at 37 °C for 20 min with Micrococcal Nuclease (#M0247S; 1:1000; New England Biolabs, Ipswich, MA) and 5 mM $CaCl_2$. Samples were diluted in lysis buffer and sonicated using a probe sonicator for $3 \times 15$ s at 30% power/50 amplitude. Supernatant was aliquoted and stored at −80 °C; DNA from one aliquot was purified after RNASe A/Proteinase K treatment to determine DNA concentration and to assess chromatin shearing.

Twenty micrograms of chromatin were incubated overnight at 4 °C with rabbit anti-Histone H3 antibody (#ab1791; 3 μg; Abcam), normal rabbit IgG (#2729; 3 μg; Cell Signaling), or rabbit anti-Smad 2/3 antibody (#D7G7; 3 μg; Cell Signaling). The immunocomplex was then mixed with 50 μl solution containing pre-blocked (with salmon sperm DNA and 1 mg/ml BSA) Dynabeads Protein A (#10002D; Thermo Fisher Scientific). Antibody-bound beads were incubated for 2 h at 4 °C on a rotator, recovered using a magnetic separation rack, and washed twice with cold 0.5 ml lysis buffer and once with cold TE buffer. Bound complexes were liberated in elution buffer (1% SDS, 100 mM NaHCO₃) twice at 60 °C for 1 h, incubated with RNASe A overnight at 55 °C, and with Proteinase K at 55 °C for 2 h. DNA was purified using the DNA Clean & Concentrator kit (Zymo Research, Irvine, CA). The purified DNA fragments were used for PCR amplification using the Phusion High-Fidelity DNA Polymerase kit (New England Biolabs) and the primers listed in Supplementary Table 4.

**Histology, immunostaining, and image analysis**. For Herovici staining, deparaffinized slides were dipped in celestine blue/iron alum solution (5 min), washed in tap water (2 min), dipped in 5% aluminum sulfate/1% alcoholic haematoxylin/4% FeCl₃/HCl solution (5 min), washed in tap water (2 min), dipped in metanil yellow/acetic acid solution (2 min), followed by acetic acid solution (2 min), washed in tap water (2 min), dipped in Li₂CO₃ solution (2 min), followed by methyl blue/acid fuchsin/picric acid/Li₂CO₃ solution (2 min), and 1% acetic acid (2 min). Picrosirius Red staining was done using the Picro Sirius Red Stain Kit (#ab150681; Abcam), and Alcian Blue with periodic acid and Schiff's solution (PAS) staining was done using the Alcian Blue PAS Stain Kit (#ab245876; Abcam), both according to the manufacturers' instructions. For staining of carboxylated/sulfated acidic glycans, the standard protocol was followed (pH 2.5), and for staining of only sulfated acidic glycans, the Alcian Blue was adjusted to pH 1.

For immunofluorescence staining of cultured cells, coverslips were fixed in 4% PFA, blocked with 1% BSA, and incubated overnight with goat anti-fibronectin (#sc-6952; 1:500; Santa Cruz, Santa Cruz, CA) or goat anti-collagen I (#1310-01; 1:500; SouthernBiotech, Birmingham, AL) primary antibodies, followed by donkey anti-goat Cy3 secondary antibody (#705-165-147; 1:1000; Jackson ImmunoResearch, West Grove, PA), and counterstained with Hoechst 33342 (Sigma). For immunofluorescence staining of co-cultures, coverslips were fixed in 4% PFA, blocked with 10% normal donkey serum (#017-000-121; Jackson ImmunoResearch), and incubated overnight with rabbit anti-fibronectin (#ab2413 1:500; Abcam, Cambridge, UK) and goat anti-collagen I (1:500; SouthernBiotech), or with rabbit anti-periostin (#ab14041, 1:200; Abcam) and goat anti-asporin primary antibodies (#ab31303, 1:200; Abcam), followed by donkey anti-goat AlexaFluor 594 (#705-585-003) and donkey anti-rabbit AlexaFluor 647 (#711-605-152) secondary antibodies (1:1000; Jackson ImmunoResearch), and counterstained with DAPI (Sigma). For immunofluorescence staining of wounded skin, 7 μm cryosections were fixed in ice-cold acetone for 10 min, blocked with 10% normal donkey serum (Jackson ImmunoResearch), and incubated overnight with rabbit anti-fibronectin and goat anti-asporin, or rabbit anti-periostin and goat anti-asporin primary antibodies (1:100; Abcam), followed by donkey anti-goat AlexaFluor 594 and donkey anti-rabbit AlexaFluor 488 secondary antibodies (1:1000; Jackson ImmunoResearch), and counterstained with Hoechst 33342 (Sigma).

Light microscopy images were acquired with the Pannoramic 250 Slide Scanner (3D Histech, Budapest, Hungary) at ×20 magnification. Polarized images were acquired with a Zeiss AxioImager.Z2 microscope (Carl Zeiss AG, Oberkochen, Germany) with a rotary stage and polarization filters at ×20 magnification. Fluorescence images were acquired with a Zeiss AxioImager.M2 microscope with a motorized stage at ×10, ×20, and ×40 magnification. For the latter two microscopes, Zen Pro software (Zeiss) was used to control the camera and to stitch together individual photomicrographs into wound- or coverslip-spanning images.

Light and polarized images were analyzed via ImagePro Plus version 4.5 (Media Cybernetics Inc., Rockville, MD). For quantification of images of Herovici- or Picrosirius Red-stained tissue sections the colors of interest were set manually. For the Herovici stain, blue pixels were considered newly deposited collagen and young, while purple pixels were considered highly crosslinked collagen and mature[17]. For the Picrosirius Red stain, green pixels were considered thin, yellow pixels were considered mixed, and red pixels were considered thick[67]. The same color segmentation settings were used for each image. For the Herovici stain, the number of stained pixels was expressed as percent of total pixels within the wound bed. For the Picrosirius Red stain, the number of stained pixels was either expressed as percentage of total pixels within the wound bed or as a percentage of total green, yellow, and red pixels. For color maps of collagen density, representative Picrosirius Red-stained histological sections were first split according to thin, mixed, and thick collagen fiber signals and then analyzed using a custom-written algorithm that allowed gradients of fiber density to be visualized. After identifying the pixels that corresponded to collagen presence, each image was split into square regions (termed elements, typical size: 50 × 50 pixel), and the relative amount of collagen per element was quantified (relevant pixels/total pixels). A smoothing step was further applied to mitigate the influence of image discretization on the determined collagen density patterns. The collagen density within each region was determined by averaging the results across neighboring elements, as defined by a centroid-to-centroid distance smaller than or equal to the length of the element's diagonal. Finally, the determined density values were normalized to the image-specific maximum and used to generate the color maps. Immunofluorescence images of co-cultures were analyzed in Fiji[68]. Stitched ×10 images of whole coverslips were cropped identically to remove edges. Individual fluorescence images were arranged in a single combined collage, identically thresholded via Otsu algorithm, and positive pixel areas were quantified in relation to the total image areas.

**RNA isolation and qRT-PCR**. RNA was isolated from cultured cells using Trizol (Life Technologies, Carlsbad, CA) or using the Mini Total RNA kit (IBI Scientific, Dubuque, IA) and reverse transcribed using the iScript cDNA synthesis kit (BioRad, Hercules, CA), all according to the manufacturers' instructions. Quantitative PCR was performed using LightCycler 480 SYBR Green I Master reaction mix (Roche, Rotkreuz, Switzerland) and the primers used in Supplementary Table 2. Data were quantified using second derivative maximum analysis and gene expression represented as relative to the internal housekeeping gene *Rps29* (for mouse samples) or *RPL27* (for human samples).

**Flow cytometric analysis and fluorescence-activated cell sorting**. Skin was excised from unwounded back skin and the underlying fat was removed. Wounds at 3-, 5-, and 7 days post-injury were excised using a 4 mm biopsy punch to exclude most of the surrounding unwounded tissue. For analysis and sorting, four 5-day wounds per mouse were pooled. Wounds and unwounded skin were stored in ice-cold RPMI-1640 medium and immediately processed. Tissue samples were minced into small pieces followed by incubation in 2 ml of medium containing 0.25 mg/ml Liberase TL (Roche), 0.25 mg/ml DNase I (Sigma), and 7.5 mM MgCl₂ for 1 h at 37°C while shaking at 100 rpm. Next, the cell suspension was passed through a 70 μm cell strainer and washed with PBS containing 0.25 mg/ml DNase I and 7.5 mM MgCl₂. Cells were centrifuged for 10 min at 1200 r.p.m., resuspended in FACS buffer (0.5% BSA, 5 mM EDTA in 1× PBS) containing FcBlock (1:200; BD BioSciences, San Jose, CA), and incubated for 10 min on ice. Cells were centrifuged for 5 min at 2000 r.p.m. and resuspended with FACS buffer containing antibodies against different cell surface markers: CD140a-APC (#135908, clone APA5, 1:200; BioLegend, San Diego, CA), CD45-Pacific blue (#103126, clone 30-F11, 1:600; BioLegend), CD11b-BV711 (#101242, clone M1/70, 1:500; Biolegend), and F4/80-PE (#123110, clone BM8, 1:200; Biolegend). After a 30 min incubation on ice, stained cells were washed with FACS buffer. To exclude non-viable cells, Sytox Green (1:1000; Invitrogen, Carlsbad, CA) was used according to the manufacturer's instructions. Cells were either analyzed using the BD LSRII Fortessa or sorted using the BD FACSAria IIIu equipped with FACSDiva software version 6 (BD Pharmingen, San Diego, CA). Fluorescence emission compensation was performed using compensation beads (BD Biosciences). Staining and gating controls included isotype control and fluorescence minus one (FMO) samples. Compensation adjustment, gating, and data analysis were performed using FlowJo software version X (Tree Star Inc, Ashland, OR) and data were exported for further processing.

For cell analysis and sorting, fibroblasts were defined as CD140a⁺ CD45⁻ CD11b⁻ F4/80⁻ live cells. The following groups were chosen for cell sorting: NS_WT, NS_Act, 5dw_WT, 5dw_Act. Four to six samples from Act and WT littermates were processed per session, and four separate sorting sessions were completed. In all, 80,000–100,000 fibroblasts per sample were sorted into RNase-free 1.5 ml tubes with 100 μl FACS buffer supplemented with RiboLock RNase Inhibitor (Thermo Fisher Scientific) at 10 U/sample. Re-analysis of sorted cells showed >90% purity of all samples.

**RNA sequencing and bioinformatics**. Sorted cells were centrifuged at 1000 g for 10 min and lysed in 350 μl RLT lysis buffer (Qiagen, Hilden, Germany) supplemented with β-mercaptoethanol (1:100). RNA from these cells was isolated via RNeasy Micro Kit (Qiagen) and their quality was analyzed using a TapeStation (Agilent Technologies, Santa Clara, CA). Samples with RIN > 6.8 were pooled from three animals across sorting sessions and adjusted to 100 ng (for NS) or 200 ng (for 5dw). Final RNA samples (n = 3 per group) were subjected to the RNA sequencing protocol via poly-A enrichment, True-Seq library preparation, and single-end 100 bp sequencing on an Illumina HiSeq 2500 v3 instrument. Sequence alignment to the mouse reference genome (build GRCm38), quality control, and initial bioinformatics analyses were performed in the CLC Genomics Workbench version 12 (Qiagen). Quality control excluded one sample in the 5dw_Act group from further analysis. All other samples generated approximately 25 million reads, showed high (82.3-95.7%) mapping rates to the mouse reference genome, high (93.7–97.5%) mapping rates to genes, and low (0.2–6.1%) rRNA mapping rates.

Numbers of RNA transcripts were determined on the gene level and normalized to reads per kilobase of transcript, per million mapped reads (RPKM). PCA was performed to evaluate grouping of samples. Genes were ranked according to absolute expression in NS and 5dw fibroblasts, and functional enrichment of the shared genes among 100 top-expressed genes was determined using ARCHS4 Cell Lines Enrichment tab in EnrichR[69] and by ImmGen MyGeneSet[70]. Differential expression analysis was performed across all group comparisons using the edgeR exact test as implemented in the CLC Genomics Workbench (Qiagen), and differentially expressed genes (DEGs) were determined and compared at the statistical significance level of false discovery rate (FDR) < 0.05. Venny (http://bioinfogp.cnb.csic.es/tools/venny/index.html) was used to compare the statistically significant (FDR < 0.05) DEGs across all 5dw vs NS comparisons. Shared up- and down-regulated DEGs were subjected to functional enrichment analysis using Gene Ontology (GO) biological processes in EnrichR. The source data file for Fig. 2 and Supplementary Fig. 2 lists the DEGs and output enrichment tables.

To find wound-regulated DEGs in fibroblasts, which are affected by activin overexpression, we first determined the NS group to use for baseline expression. We compared DEGs (FDR < 0.05), which are at least 5% greater in the following two comparisons: (1) log₂(5dw_Act/NS_Act) vs log₂(5dw_WT/NS_WT); (2) log₂(5dw_Act/NS_WT) vs log₂(5dw_WT/NS_WT). Venny was used to compare the genes, which found that a majority are shared between the two comparisons, and functional enrichment of the shared genes was done using the Reactome database in EnrichR. The Matrisome database[31] was used to extract the ECM-regulated genes in these comparisons. We selected NS_WT as the baseline condition, and used comparison #2 to generate a ranked list of activin-regulated, wound-expressed DEGs in fibroblasts. The source data file for Fig. 3 and Supplementary Fig. 3 lists the activin-regulated genes and Matrisome-filtered genes.

**Ingenuity pathway analysis**. Wound-regulated genes (556 DEGs) and activin-regulated genes (165 DEGs), in conjunction with their Log₂Ratio and FDR values, were uploaded to IPA (Qiagen). Core analyses were performed using confidence of only experimentally observed relationships. Data tables associated with Diseases & Functions were exported to identify highly activated enriched functions and regulators in generic wound and activin-exposed wound fibroblasts. Selected functions were arranged in a spreadsheet, and their activation Z-scores were color-coded for visualization. Original IPA output data with unfiltered Diseases & Functions for all comparisons are provided in the source data files for Figs. 2 and 3.

**Gene set enrichment analysis**. Sets of significantly up- and down-regulated genes were generated by mining of various relevant databases, including SEEK for co-expressed genes, GEO for published transcriptomic data of wound-derived (myo) fibroblasts, skin wound healing, keloid and other scar-forming fibroblasts, and published single-cell RNA-seq data. Original gene sets were uploaded to GSEA version 3 and filtered to those mapped by gene symbol and present in the tested datasets. The gene sets were tested against the generic wound fibroblast signature (5dw vs NS) and the pre-ranked list of activin-regulated wound fibroblast DEGs. GSEA results were organized in tables, and the normalized enrichment scores (NES) and false discovery rate (FDR) values were color coded for visualization. Leading Edge analyses were extracted to identify commonly enriched genes between gene sets. The previously published datasets and samples analyzed, original and filtered gene sets, GSEA settings used for each analysis, ranked gene lists, and original GSEA output data, including leading edge analyses for all experiments, are provided in the GSEA-specific source data file for Figs. 2 and 3.

**Collagen and collagen cross-link analysis**. For analysis of collagen and of collagen cross-links[51], normal skin and wound samples were treated with sodium borohydride (Sigma, 25 mg NaBH$_4$/ml in 0.05 M NaH$_2$PO$_4$/0.15 M NaCl pH 7.4, 1 h on ice, 1.5 h at RT) to stabilize reducible acid-labile cross-links, digested for 12 h at 37 °C with high purity bacterial collagenase (C0773; Sigma, 50 U/ml), and hydrolyzed in 6 N HCl at 110 °C for 24 h. The hydrolysates were precleared by solid phase extraction and analyzed on an amino acid analyzer (Biochrome30, Biochrome, Cambridge, UK). Quantification was based on ninhydrin-generated leucine equivalence factors (DHLNL, HLNL, and HP: 1.8; HHMD: 3.4). The nomenclature used in the manuscript refers to the reduced variants of cross-links (DHLNL, HLNL, HHMD). The ratio of the number of hydroxylysine-aldehydes to lysine-aldehydes involved in cross-links was calculated as $(2 \times HP + DHLNL)/(2 \times HHMD + HLNL)$. HLNL was grouped as lysine-aldehyde-derived cross-link, because it has been shown that the helical lysine residues utilized for cross-linking (K87, K930, K933) are hydroxylated in skin[71]. Therefore, HLNL forms between telopeptidyl lysine aldehyde and helical hydroxylysine. Collagen and protein contents of the specimens were analyzed in an aliquot of hydrolyzed specimens of the soluble and the residual fraction after collagenase solubilization prior to solid phase extraction. Collagen content was calculated based on a content of 14 mg hydroxyproline in 100 mg collagen.

**Non-invasive optical assessment of wound morphology and deformability**. For non-invasive determination of the macroscopic, morphological, and biomechanical wound characteristics over the time course of healing, we developed a measurement technique akin to that presented in ref. [72], and extended its use to murine skin wounds. The wounded mice were anaesthetized using isoflurane and laid on an operating stage, which was warmed to 37 °C (Supplementary Fig. 8a). Following hair clipping and disinfection with ethanol, two strips of medical-grade tape (Leukotape; BSN Medical GmbH, Hamburg, Germany) were applied roughly 10 mm proximally and distally to the wound, and used to extend it in the cranio-caudal direction (Supplementary Fig. 8a). The chosen loading level was nearly physiological and thus did not damage the healing or healthy tissue. To enhance measurement safety, the tape bands were pulled manually in order to provide immediate feedback on the applied load level and thus enhance measurement safety. This method does not provide direct control over the applied strain rate, and post hoc analysis showed that the applied rates were comprised between 0.044 and 0.230 s$^{-1}$, with an average value of 0.094 s$^{-1}$ and an SEM of 0.002 s$^{-1}$. However, the influence of the loading rate variability on the present quantification of relative tissue deformability in wounded skin was shown to be very modest (see below, Supplementary Fig. 9a).

The deformation of the extended skin region was monitored by an optical setup, comprising a charge-coupled-device (CCD) camera (UI-2220SE-M-GL; IDS Imaging Development Systems GmbH, Obersulm, Germany) and a ×0.25 telecentric lens (NT55-349; Edmund Optics GmbH, Karlsruhe, Germany) providing a field of view of $30 \times 30$ mm$^2$ with a $1000 \times 1000$ pixel size; the use of telecentric lenses is essential due to the variability of the relative position between skin surface and objective during mechanical loading. The image acquisition was controlled by a laptop computer via the camera software, allowing video sequences of the tissue extension to be recorded and stored for subsequent analyses. This provided for simultaneous monitoring of the evolution of: (i) the wound morphological extension; (ii) the skin deformability in and around the wound; and (iii) the extension of the region mechanically affected by the wound.

For each video sequence, the first recorded image frame was used to measure the morphological wound extension by using ImageJ software (National Institutes of Health, Bethesda, MD) to manually fit an ellipse to the visible wound region. Quantification of the corresponding area and length along the cranio-caudal direction provided information on the extent of wound closure/contraction throughout healing, including the extent of scar formation (Supplementary Fig. 8b, d).

The recorded image sequences additionally allowed the heterogeneous field of tissue deformation in the skin region of interest to be reconstructed. We first selected fiducial points within a wide tissue region including the wound (cf. green dots in Supplementary Fig. 8f) and tracked their displacement upon extension. Point identification and tracking was based on the contrast pattern naturally offered by the shaved and wounded skin, according to the custom-written Python-based algorithm (Python Software Foundation, Wilmington, DE) that was previously presented for applications to homogeneous deformation fields[73,74]. The

displacement vector of each tracking point was first used to determine the global level of mechanical strain along the direction of tissue extension, corresponding to the slope of the regression line, which best described (in a least-square sense) the relationship between point displacements and reference positions[17]. This provided for standardization across all measurements, allowing identification of the instant at which the prescribed 10% (or 5%) global strain level was reached. Then, the inhomogeneous deformation field within and around the wound was quantified by determining the level of strain at each tracking point location. This was attained by estimating the components of the corresponding 2D deformation gradient tensor ($\mathbf{F} = \frac{\partial \mathbf{x}}{\partial \mathbf{X}}$, $\mathbf{x}$ and $\mathbf{X}$ being the 2D position vectors in the deformed and reference configuration, respectively) using weighted least-square fitting. Letting $P_j$ be the location where the strain is to be determined, $P_i$ the $i$th tracking point, and $\|P_i - P_j\|$ their mutual distance, the weights were given by $w_{ij} = (\|P_i - P_j\| + 1)^{-1}$ if $\|P_i - P_j\| < R_{max}$ and $w_{ij} = 0$ otherwise; based on the typical tracking pattern offered by the shaved skin, the value of $R_{max}$ was here set to 100 pixels. The (Green–Lagrange) strain tensor at each tracking point was then computed as $\mathbf{E} = \frac{1}{2}(\mathbf{F}^{\mathrm{T}}\mathbf{F} - \mathbf{I})$, $\mathbf{I}$ being the $2 \times 2$ identity matrix, and its component along the cranio-caudal direction was used to generate color maps of tissue deformability. To provide a metric for comparison of wound healing progression, a region within the visible wound area was selected and the corresponding average strain was quantified (cf. black box, wound area in Supplementary Fig. 8f).

The recorded image sequences additionally allowed the measure the extension of the tissue region that is biomechanically affected by the wound presence, thus providing a metric for wound size that also relates to tissue functional recovery and not simply to wound morphology. We further focused the analysis on a narrow tissue strip including the wound and generated a displacement vs reference position chart. Least-square-based determination of regression lines for the regions located within and around the wound[17] showed that the biomechanically affected tissue most often extended beyond the morphologically visible wound areas (cf. Supplementary Fig. 8e); the corresponding distance was thus termed functional wound length and quantified throughout the experimental campaign.

The dynamics of macroscopic tissue biomechanical and morphological changes was also assessed by determining the rate of change of the parameters presented above. The rate of change in wound length and deformability at days 5–14 were approximated according to a central finite-differentiation scheme, accounting for the uneven spacing between measurement time points:

$$\frac{\partial f_i}{\partial t_i} \approx \frac{t_i - t_{i-1}}{[(t_{i+1} - t_i)(t_{i+1} - t_{i-1})]} f_{i+1} + \frac{t_{i+1} - 2t_i + t_{i-1}}{[(t_{i+1} - t_i)(t_i - t_{i-1})]} f_i - \frac{t_{i+1} - t_i}{[(t_i - t_{i-1})(t_{i+1} - t_{i-1})]} f_{i-1}, \quad (1)$$

where $f$ is the function whose rate is to be determined, $t_i$ the time point at which the rate is computed, $t_{i-1}$ the previous time point, and $t_{i+1}$ the subsequent one. Conversely, forward differentiation was used for the day 3 rates, $\frac{\partial f_{3d}}{\partial t_{3d}} \approx \frac{f_{5d} - f_{3d}}{t_{5d} - t_{3d}}$, and backward differentiation for the day 21 rates, $\frac{\partial f_{21d}}{\partial t_{21d}} \approx \frac{f_{21d} - f_{14d}}{t_{21d} - t_{14d}}$.

In particular, the rates of wound length change were computed both starting from average and individual values of wound length. While the former metric provides an overview of the wound healing dynamics, the latter allows for direct statistical comparison of rates across time points and animal phenotypes. However, the latter was not consistently available throughout the experimental campaign due to the variable number of wounds tested at each time point (cf. legends of Fig. 8b vs c, right panel).

**Assessments of robustness and reproducibility of wound deformability measurements**. The choice of the 10% global strain did not affect the measured progression of the skin biomechanical properties, as shown by comparing results obtained at a global strain of 5% or 10% (Supplementary Fig. 9a). Furthermore, varying the applied loading rate by 100-fold marginally altered the elastograms and the quantified wound strains in a synthetic model system (Supplementary Fig. 9b–d) and in a non-linear rate-dependent finite-element simulation (Supplementary Fig. 9e–g).

Due to the chosen manual method for load application, the imposed loading rate might vary considerably in the different experiments. To assess if these variations affect the results, we used a rate-dependent synthetic model system of a cutaneous wound. This was obtained by inserting a 6-mm-diameter disc of sandpaper between two sheets of the acrylic elastomer VHB 4910$^{\mathrm{TM}}$ (3M GmbH, Rüschlikon, Switzerland), whose time-dependent mechanical characteristics have been previously quantified[75]. Following surface preparation for optical strain tracking[74], the specimen (clamp to clamp distance: 40 mm, width: 50 mm) containing the stiff inclusion at its center was mounted on our previously described custom-made setup for mechanical testing[74] and extended to about 15% nominal strain at a strain rate of either 0.10 or 0.001 s$^{-1}$. The former value was chosen to be representative of the strain rate applied in the in vivo experiments, whereas the latter was chosen to represent a particularly slow elongation. As a consequence of the viscoelastic behavior of elastomer, the force required for the low strain rate extension was about one half of that corresponding to the faster loading (Supplementary Fig. 9b). The elastograms obtained using the algorithm described in our manuscript displayed only modest differences between the two cases, despite

the large strain rate variation applied (Supplementary Fig. 9c). Quantifying the strain in the center of the stiff inclusion (Supplementary Fig. 9d) further showed a limited increase (approximately 2.2% vs 1.7%) despite the extremely large, namely 100-fold, variation in the imposed strain rate. Note that this difference is much lower than the relative variations observed in the course of healing (Fig. 8g).

We further corroborated these experimental results by performing non-linear rate-dependent finite-element simulations of load application to a wounded skin sample with planar dimensions of $25 \times 25$ mm$^2$, thickness of 1 mm, and including a 5-mm-diameter wound at its center. The constitutive behavior of healthy skin tissue was directly adopted from refs. [76,77], where a model of skin's time-dependent mechanical behavior in vivo was derived and validated; the parameter $\mu_0$, akin to a shear modulus for the tissue, was increased by one order of magnitude in order to represent the wounded tissue behavior. Elongating the specimen by 10% in either 100 or 1 s yielded again strain rates differing by a factor of 100, which resulted in an almost two-fold increase in the reaction force (Supplementary Fig. 9e). This is similar to the experimental results obtained on a specimen of VHB 4910$^{TM}$ (Supplementary Fig. 9b). Despite the important variation in the imposed strain rate, the predicted strain patterns obtained from finite-element simulations are not visibly different inside or outside of the stiff inclusion (Supplementary Fig. 9f). Quantifying the strain level at the center of the wounded region confirmed the modest influence of strain rate on this parameter (Supplementary Fig. 9g).

The reproducibility of deformability measurements was evaluated through a comparison of data from WT mice (Supplementary Fig. 9h). Permutations of two independently performed experiments showed significant differences between experiments at d3 vs d10 (Supplementary Fig. 9h top). By contrast, there were no statistically significant differences for independent measurements corresponding to the same time points at d3, d10, and d14 (Supplementary Fig. 9h bottom).

**Mechanical testing of excised tissue specimens**. Twenty-one-day-old wounds were excised after sacrifice of the mice, and specimens containing the scar were subjected to mechanical testing under uniaxial tensile conditions[17]. Compared to the in vivo methodology developed here, this configuration provides easier control of the boundary and testing conditions as well as quantitative insight into tissue stiffness, at the expenses of being far from physiological in vivo conditions. To test the scar mechanical behavior in a physiologically relevant range of deformations, we focused our analysis on the instant when a strain level of 10% was reached in the tissue far away from the wound. The corresponding scar deformation was extracted and compared for both mouse genotypes, using wounds from WT and Act mice. In line with the in vivo measurements, the strain was here defined as $\varepsilon = (\lambda_1^2 - 1)/2$, $\lambda_1$ being the tissue stretch along the testing direction. Additionally, the ultimate nominal tension (i.e. force/reference width), corresponding to the moment when a fissure was visible, was also quantified according to ref. [17] in order to provide a measure of scar tissue strength.

To compare the non-wounded skin of WT and Act mice, and possibly identify characteristic biomechanical differences that do not arise as part of the wound healing process, we additionally subjected tissue specimens (gauge dimensions: $20 \times 5$ mm$^2$, length × width) to a uniaxial testing protocol[73], and quantified their resistance to deformation as well as the in-plane kinematics[73].

The thickness of skin from Act and WT mice was quantified by leveraging a custom-built setup for confined compression tests, which allowed determining the resting position of a shaft weighing 20–30 g and placed in contact with the tissue.

**Statistics**. Statistical analysis was performed using the PRISM software, version 8 for Mac OS X or Windows (GraphPad Software Inc., San Diego, CA). Shapiro–Wilk test was performed to validate normality of data prior to statistical testing. For comparison of two groups, $F$ test was used to verify equal variance between groups, followed by the two-sided Student's $t$-test; for comparison of more than two groups, Brown–Forsythe test was used to verify equal variance between groups, followed by one-way or two-way ANOVA and Tukey's or Bonferroni's multiple comparisons post hoc tests. Equivalence tests using 90% confidence intervals were conducted[78]. Exact $P$ values of statistical comparisons are shown in the figures.

**Reporting summary**. Further information on research design is available in the Nature Research Reporting Summary linked to this article.

## Data availability

The authors declare that all data supporting the findings of this study are available within the article, its supplementary information file, in the source data files or from the corresponding authors upon reasonable request. RNA sequencing data from skin and wound fibroblasts of WT and Act mice have been deposited in the Gene Expression Omnibus database under accession code: GSE134789. Source data files have been deposited in the ETH Zurich Research Collection (https://doi.org/10.3929/ethz-b-000409545 [https://www.research-collection.ethz.ch/handle/20.500.11850/409545]).

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

## Acknowledgements

We thank Dr. Eric Haertel, Chelsea Chen, Asli Adak, Nicolas Mathis, and Luca Ferrarese (Institute of Molecular Health Sciences, ETH Zurich) for invaluable experimental help, Sol Taguinod (ETH Phenomics Center) for help with the mouse maintenance, Dr. Malgorzata Kisielow and Anette Schütz (ETH Flow Cytometry Core Facility) for help with the flow cytometry experiments, Catharine Aquino (Functional Genomics Center Zurich) for performing the RNA-sequencing experiments, Dr. Raoul Hopf (Institute for Mechanical Systems, ETH Zurich) for consulting on optical strain analysis, Dr. Hans-Dietmar Beer (University of Zurich) for providing human primary fibroblasts, and Dr. Petra Boukamp (Leibniz Research Institute for Environmental Medicine, Düsseldorf, Germany) for providing HaCaT keratinocytes. This work was supported by grants from the Swiss National Science Foundation (205321_179012 to E.M and 31003A_169204 to S.W.), from Cancer Research Switzerland (KFS-4510-08-2018 to S.W.), and from University Medicine Zurich, Flagship Project SKINTEGRITY (to E.M and S.W.).

## Author contributions

S.W. and E.M. designed the study together with M.S.W. and M.P., and supervised the work. M.S.W., M.P., M.C., B.M., J.J., and J.B. performed experiments and analyzed data. M.S.W. and M.P. made the figures. M.S.W., M.P., and S.W. wrote the manuscript. S.W. and E.M. provided the funding. All authors made important suggestions to the manuscript.

## Competing interests

The authors declare no competing interests.

**Additional information**

