## [Peer review file · Nature Communications]

Reviewers' Comments:

Reviewer #1:

Remarks to the Author:

In this study authors aim to comprehensively examine gene expression, ECM and biomechanical properties of mouse skin wounds and how they are affected by Activin. Authors conclude that Activin alters wound healing at multiple levels and that the combined effect of these alterations is more prominent fibrosis (as summarized in Supplemental Fig. 7).

The study has strong points. Specifically:

1) Authors do a good job comparing their own RNA-seq data to multiple other published RNA-seq datasets on wound fibroblasts. To my knowledge, this is the first such comprehensive meta-analysis. As the result of this analysis they identify a core gene expression signature of wound fibroblasts, that features a large number of ECM proteins and ECM-modifying enzymes. This is a good example of a comprehensive genomic data meta-analysis.

2) Authors propose new non-invasive method for measuring wound tissue deformability. In principle, it can be useful. Authors describe technical aspects of the assay in the supplement in what appears to be sufficient details for reproducibility.

At the same time, authors data on the wound healing phenotype of Activin over-expressing mice is for the most part descriptive and correlative and not sufficiently mechanistic. Specific weaknesses include:

3) The mutant mouse model employed by authors is K14-Activin. These mice over-express Activin in the epithelial cells. In the wound, the primary source of transgenic Activin will be wound epidermis. Yet, the key phenotype - changes in the forming dermal scar tissue are seen throughout the wound, even in its deepest layers - at the morphological distances that are not consistent with the morphogenetic gradients of most soluble growth factors in tissues, which is typically on the order of 50-100 μ m. Therefore, it is very likely that many dermal scar changes observed by authors in K14-Activin mice, including gene expression changes in wound fibroblasts, are not the direct effect of increased Activin signaling in fibroblasts. There can be additional intermediate signaling changes that directly affect deep layer fibroblasts, that are not accounted for.

Authors performed a limited set of in vitro studies on early postnatal (P2.5) mouse skin fibroblasts and human foreskin fibroblasts in the presence of human recombinant Activin and showed by qPCR that several genes differentially expressed in their wound fibroblast RNA-seq data also become over-expressed in vitro. This is encouraging. However, these fibroblasts are different from adult mouse wound fibroblasts, the qPCR assay is done in a much simpler in vitro environment and they do not definitively establish selected genes as direct targets of Activin pathway with relevance to observed wound healing phenotypes.

More detailed, genetic and preferably in vivo studies will be needed to identify and confirm direct targets of Activin signaling in wound fibroblasts. These experiments can include, for example, genetic rescue studies -- such as fibroblasts-specific mutations to Activin receptors or intracellular mediator(s) of Activin pathway.

4) Authors mention it, but do not specifically examine, the fact that wound fibroblasts also produce Activin normally. Can it act in a positive feedback loop and can the introduction of transgenic Activin from K14+ epidermis become amplified through the wound dermis in a positive feedback loop type of mechanism? If this is a likely possibility, it needs to be tested.

5) One of the most obvious phenotypes seen in wounds of K14-Activin mice is prominently faster wound re-epithelialization. As it is obvious in main Fig. 1, K14-Activin wounds are already fully re-

epithelialized by day 5 and their wound epidermis remains prominently hyperplastic even on day 14 (while in control mice, epidermal hyperplasia subsides).

Therefore, it appears that the leading direct phenotype of Activin overexpression is the epidermal phenotype. Dermal differences can arise from the faster epidermis repair kinetics, since wound epidermis is a known prominent source of signaling and its signaling outputs will be different between fully re-epithelialized state and actively re-epithelializing states. Ideally, fibroblast-specific Activin pathway mutation (rather than epidermal) needs to be employed. This will require a new type of transgenic mouse or some other form of dermis-specific Activin delivery.

Reviewer #2:

Remarks to the Author:

The manuscript by Wietecha et al., revisit their previously published activin-overexpression mouse model that they have used for wound healing studies. Recapitulating their own previous results, they show that Activin overexpression results in accelerated wound healing and increased granulation tissue formation. They then perform RNAseq analyses, comparing non-wounded WT and Activin-overexpressor FACS purified fibroblasts with wounded WT and overexpressing fibroblasts. As would be expected, wounding induces substantial changes in fibroblast transcriptomes in both genotypes. More surprisingly, no significant changes are observed in response to Activin overexpression (1-4 differentially expressed genes with a false discovery rate of 5%). Importantly, PCA plots in Supplementary Fig. 2A show complete interspersation of the two genotype samples indicative of their extreme similarity and more variation being present in the individual samples than the experimental groups. Nevertheless, the authors proceed to make complex comparisons of the transcriptome data to identify "activin regulated genes" by comparing individual read counts of genes (% increase in gene expression with a 5% increase being a cutoff) in the dataset. This is my main concern of the manuscript: the title implies that "Activin reprograms the fibroblast transcriptome". The data does not support such conclusions as no significant differences in transcriptome results from activin overexpression and the biological effects of the gene expression changes in the wounds in terms of fold changes are extremely small. These types of marginal changes in RNA are unlikely to result in biologically significant protein-level changes; The increased granulation tissue and altered mechanical properties could be explained purely with increased numbers of wound fibroblasts, which would subsequently lead to increased matrix production and altered granulation tissue. This alternative explanation, in particular in the light of the non-existent transcriptional changes should be addressed. In addition, the title should be modified to more appropriately describe the findings that implicate lack of large scale changes in the transcriptome, and the conclusions should be toned down throughout.

Other points:

1. The measurement of wound strain is interesting and potentially very relevant, but seems to be based on manual pulling of medical tape, which could lead to precision problems due to variable and non-controlled loading rates of the strain. It would be important to see a calibration and reproducibility curves of this method for example using sheets of viscoelastic materials of known properties.

Response to Reviewers

Reviewer #1 (Remarks to the Author):

In this study authors aim to comprehensively examine gene expression, ECM and biomechanical properties of mouse skin wounds and how they are affected by Activin. Authors conclude that Activin alters wound healing at multiple levels and that the combined effect of these alterations is more prominent fibrosis (as summarized in Supplemental Fig. 7)

The study has strong points. Specifically:

1) Authors do a good job comparing their own RNA-seq data to multiple other published RNA-seq datasets on wound fibroblasts. To my knowledge, this is the first such comprehensive meta-analysis. As the result of this analysis they identify a core gene expression signature of wound fibroblasts, that features a large number of ECM proteins and ECM-modifying enzymes. This is a good example of a comprehensive genomic data meta-analysis.

2) Authors propose new non-invasive method for measuring wound tissue deformability. In principle, it can be useful. Authors describe technical aspects of the assay in the supplement in what appears to be sufficient details for reproducibility.

Our reply: We thank the reviewer for the positive comments!

At the same time, authors data on the wound healing phenotype of Activin over-expressing mice is for the most part descriptive and correlative and not sufficiently mechanistic. Specific weaknesses include:

3) The mutant mouse model employed by authors is K14-Activin. These mice over-express Activin in the epithelial cells. In the wound, the primary source of transgenic Activin will be wound epidermis. Yet, the key phenotype - changes in the forming dermal scar tissue are seen throughout the wound, even in its deepest layers - at the morphological distances that are not consistent with the morphogenetic gradients of most soluble growth factors in tissues, which is typically on the order of 50-100 um. Therefore, it is very likely that many dermal scar changes observed by authors in K14-Activin mice, including gene expression changes in wound fibroblasts, are not the direct effect of increased Activin signaling in fibroblasts. There can be additional intermediate signaling changes that directly affect deep layer fibroblasts, that are not accounted for.

Our reply: This is an important point and we noticed that we had not sufficiently explained this issue in the manuscript. In contrast to most soluble growth factors that mainly act in an autocrine or paracrine manner, activin A is an endocrine acting hormone, which is highly diffusible. For example, it determines cell fate during mesoderm induction or neurulation in *Xenopus laevis* embryos throughout a distance of several cell diameters and acts as a classical morphogen. We now mention this in the Introduction and we cite the relevant literature (Jones et al., 1996, PMID 8939607; McDowell et al., 1997, PMID 9285724; Shimizu and Gordon, 1999, PMID 10359791). The high diffusibility of activin A is also reflected by the fact that activin A serum levels are strongly elevated in the K14-Activin mice (Antsiferova et al., 2011, PMID 22146395). Thus, the activin A produced by keratinocytes must diffuse through the dermis to the vessels and therefore,

at least the most upper fibroblasts should be reached. In addition, the signal is amplified by auto-induction of activin A in fibroblasts (see below). We have clarified this issue in the revised version of the manuscript (page 4). Nevertheless, based on the epidermal overexpression we expect a particularly strong effect of activin on the fibroblasts that are close to the wound epidermis or injured hair follicles. Indeed, immunofluorescence staining for the activin targets asporin and periostin was most prominent at the wound edge and thus below the activin A-producing hyperproliferative wound epidermis (see Fig. 5b-e and Supplementary Fig. 5c); we also mention this result in the text (page 15, first paragraph). Importantly, our new data from keratinocyte-fibroblast co-culture experiments (explained below) show the strong induction of ECM production by fibroblasts in the immediate vicinity of keratinocytes, particularly of activin A-overexpressing keratinocytes (see Fig. 4e, and additional details in Supplementary Fig. 4c); we also mention this observation in the text (page 14, first paragraph).

Furthermore, the activin A-stimulated fibrosis “signature” was most pronounced in the wound fibroblast clusters, which were found to localize to the upper granulation tissue (cluster 8) of large day 12 wounds or which localized throughout the granulation tissue of these wounds (cluster 2). We now mention this in the text (see sc-RNAseq comparison (page 11, second paragraph; Fig. 3a iv) and reference Guerrero-Juarez et al., 2019, PMID 30737373).

Authors performed a limited set of in vitro studies on early postnatal (P2.5) mouse skin fibroblasts and human foreskin fibroblasts in the presence of human recombinant Activin and showed by qPCR that several genes differentially expressed in their wound fibroblast RNA-seq data also become over-expressed in vitro. This is encouraging. However, these fibroblasts are different from adult mouse wound fibroblasts, the qPCR assay is done in a much simpler in vitro environment and they do not definitively establish selected genes as direct targets of Activin pathway with relevance to observed wound healing phenotypes.

Our reply: We addressed this important concern in the following ways:

- a.) We tested if some of the genes that are regulated by activin A *in vivo* are direct targets of activin signaling. Indeed, we identified multiple Smad2/3 binding elements in the promoter or in the first intron of the *Postn* and *Aspn* genes (selected as examples for activin targets), of which some are highly conserved among different species (see Supplementary Fig. 4b). Chromatin immunoprecipitation (ChIP) experiments demonstrate that Smad2/3 indeed bind to these elements in response to activin A treatment, thus identifying *Aspn* and *Postn* as direct activin targets. We used immortalized fibroblasts from young mice for this purpose, since we need a large number of cells for these experiments and primary fibroblasts from adult mice grow poorly and can only be passaged once. These new results are now shown in Fig. 4c.
- b.) We established fibroblasts from 5-day wounds, starved them to reduce the effect of the TGF- β in the medium, and treated them with recombinant activin A. qRT-PCR experiments revealed that recombinant activin A also induces expression of *Inhba* and *Wisp1* in wound fibroblasts. Interestingly, expression levels of *Col1a1* and *Aspn* were already significantly higher in cultured wound fibroblasts of Act vs Wt mice, and recombinant activin A did not further induce their expression at the time point that we analyzed. These findings, shown in Fig. 4b, suggest that the wound fibroblasts from Act mice are already activated and that resulting epigenetic alterations may be maintained *in vitro*. Indeed, expression of *Col1a1* and *Lox* was significantly higher in fibroblasts from 5-day wounds compared to fibroblasts from non-injured skin of mice at the same age. These new results are now shown in Supplementary Fig. 4a.

c.) We generated immortalized human keratinocyte lines (HaCaT cells) with doxycycline (Dox)-inducible activin A overexpression and used them in 2D co-cultures with immortalized fibroblasts with Dox-inducible expression of a dominant-negative activin receptor IB mutant (dnActRIB). FACS isolation of keratinocytes and fibroblasts from these co-cultures and subsequent qRT-PCR analysis revealed that activin A overexpression in keratinocytes promotes the expression of *Aspn* in fibroblasts of the co-culture. Importantly, the increase in *Aspn* expression was abolished when fibroblasts expressing dnActRIB were used for the co-culture. *Lox* and *Postn* expression was also strongly suppressed in fibroblasts of dnActRIB mice, indicating that their expression is dependent on activin receptor signaling. These new data are shown in Fig. 4f. Furthermore, we show by immunofluorescence analysis that activin A promotes the deposition of collagen type I, fibronectin, asporin and periostin in the ECM of these co-cultures, in particular in the vicinity of the activin-overexpressing keratinocytes. These effects were again abrogated when keratinocytes were co-cultured with fibroblasts expressing dnActRIB. These new results are now shown in Fig. 4e and Supplementary Fig. 4c.

More detailed, genetic and preferably in vivo studies will be needed to identify and confirm direct targets of Activin signaling in wound fibroblasts. These experiments can include, for example, genetic rescue studies -- such as fibroblasts-specific mutations to Activin receptors or intracellular mediator(s) of Activin pathway.

Our reply: As mentioned above, we now identified *Aspn* and *Postn* as direct targets of activin A using chromatin immunoprecipitation and qRT-PCR analysis after *in vitro* short-term treatment of skin fibroblasts with recombinant activin A. The same genes/proteins are also upregulated in the K14-Act mice *in vivo*. Since these genes are mainly or even exclusively expressed by fibroblasts and since they were upregulated in isolated wound fibroblasts and also in activin A-treated cultured fibroblasts, these cells are obvious targets of activin signaling in the wound tissue. We have now further clarified this issue in the text (pages 12-13).

Furthermore, and as mentioned above, the results obtained from the 2D co-culture experiments using activin-overexpressing immortalized human keratinocytes and fibroblasts expressing dnActRIB further confirm that several ECM genes are targets of activin A in fibroblasts. Thus, we successfully performed a genetic rescue *in vitro*. A similar experiment *in vivo* would require generation of new cell-type specific transgenic mice (mice expressing dnActRIB in fibroblasts) and their mating with Act mice. This is unfortunately not possible within the time frame allowed for the revision, but we will certainly consider this important experiment for future studies.

4) Authors mention it, but do not specifically examine, the fact that wound fibroblasts also produce Activin normally. Can it act in a positive feedback loop and can the introduction of transgenic Activin from K14+ epidermis become amplified through the wound dermis in a positive feedback loop type of mechanism? If this is a likely possibility, it needs to be tested.

Our reply: Yes, there is indeed a positive feedback loop, which we now show in our qRT-PCR experiments using wound fibroblasts treated with activin A (Fig. 4b), and we mention this explicitly in the text (first paragraph of page 13). The result is also consistent with recent findings from our group showing activin A autoinduction in skin fibroblasts in a tumor setting (Cangkrama et al., 2020; in press) and with older findings in *Xenopus* embryos (Suzuki et al., 1994; PMID 8135731).

5) One of the most obvious phenotypes seen in wounds of K14-Activin mice is prominently faster

wound re-epithelialization. As it is obvious in main Fig. 1, K14-Activin wounds are already fully re-epithelialized by day 5 and their wound epidermis remains prominently hyperplastic even on day 14 (while in control mice, epidermal hyperplasia subsides).

Therefore, it appears that the leading direct phenotype of Activin overexpression is the epidermal phenotype. Dermal differences can arise from the faster epidermis repair kinetics, since wound epidermis is a known prominent source of signaling and its signaling outputs will be different between fully re-epithelialized state and actively re-epithelializing states. Ideally, fibroblast-specific Activin pathway mutation (rather than epidermal) needs to be employed. This will require a new type of transgenic mouse or some other form of dermis-specific Activin delivery.

Our reply: We now further addressed this issue by 2D co-culture experiments that involve a genetic rescue, as explained above. We would also like to point out that activin A does not directly promote, but rather inhibits keratinocyte proliferation (Seishima et al., 1999; PMID 10201525). Therefore, the strong effect of keratinocyte-derived activin A on wound healing is mainly mediated via the dermis/granulation tissue (see Introduction, page 4). This is consistent with our previous data obtained in a cancer setting, where activin was shown to exert a pro-tumorigenic effect via the stroma (Antsiferova et al., 2011). We agree that an effect of the more advanced wound epidermis on the *in vivo* wound healing phenotype cannot be excluded, but the set of new data that we obtained clearly point to a direct effect of activin A on fibroblasts and resulting matrix deposition. We now discuss this important point in the Discussion (page 21, last paragraph and page 22, first paragraph).

Reviewer #2 (Remarks to the Author):

The manuscript by Wietecha et al., revisit their previously published activin-overexpression mouse model that they have used for wound healing studies. Recapitulating their own previous results, they show that Activin overexpression results in accelerated wound healing and increased granulation tissue formation. They then perform RNAseq analyses, comparing non-wounded WT and Activin-overexpressor FACS purified fibroblasts with wounded WT and overexpressing fibroblasts. As would be expected, wounding induces substantial changes in fibroblast transcriptomes in both genotypes. More surprisingly, no significant changes are observed in response to Activin overexpression (1-4 differentially expressed genes with a false discovery rate of 5%). Importantly, PCA plots in Supplementary Fig. 2A show complete interspersions of the two genotype samples indicative of their extreme similarity and more variation being present in the individual samples than the experimental groups. Nevertheless, the authors proceed to make complex comparisons of the transcriptome data to identify “activin regulated genes” by comparing individual read counts of genes (% increase in gene expression with a 5% increase being a cutoff) in the dataset. This is my main concern of the manuscript: the title implies that “Activin reprograms the fibroblast transcriptome”. The data does not support such conclusions as no significant differences in transcriptome results from activin overexpression and the biological effects of the gene expression changes in the wounds in terms of fold changes are extremely small. These types of marginal changes in RNA are unlikely to result in biologically significant protein-level changes; The increased granulation tissue and altered mechanical properties could be explained purely with increased numbers of wound fibroblasts, which would subsequently lead to increased matrix production and altered granulation tissue. This alternative explanation, in particular in the light of the non-existent transcriptional changes should be addressed. In addition, the title should be modified to more appropriately describe the findings that implicate lack of large scale changes in the transcriptome, and the conclusions should be toned down throughout.

Our reply: This is a very important point and we had obviously not sufficiently explained this issue in the initial version of our manuscript. We have now addressed this issue as follows:

We agree that the differences between genotypes for the 5-day wound fibroblast transcriptomes are rather minor. However, the extent of regulation by activin A in the RNA-seq data is most likely underestimated, since we used all fibroblasts from the wound and the wound edge for the analysis. Importantly, the cells that are close to the activin A-overexpressing keratinocytes are more affected by the epidermal overexpression than other cells, especially at day 5 post-injury (see below). We also want to point out that in the RNA-seq data the differences between genotypes are small compared to the massive changes between unwounded and wounded skin and are actually masked by them in the original PCA plot. Indeed, when we re-do the PCA plot on only the 5-day wound fibroblast transcriptomes, we see separation between the genotypes, particularly along PC2 and PC3, although inter-sample variability is still quite high, probably due to the inherent heterogeneity of the bulk-sequenced sorted wound fibroblasts. This result is now shown in Supplementary Fig. 2d.

As explained in the response to reviewer #1, the cells that are close to the activin A-overexpressing keratinocytes are more affected by the epidermal overexpression than other cells. Therefore, the results obtained with the bulk fibroblast population is most likely underestimated. This is demonstrated by the particularly strong increase in asporin and periostin expression that we observed at the wound edge in the K14-Act mice (Fig. 5b-c; and Supplementary Fig. 5b). This finding also demonstrates that the observed small increase in the mRNA levels of these genes translates into differences at the protein level. We also validated some of the most strongly activin-regulated genes identified *in vivo* using cultured fibroblasts (see below).

The alternate explanation for phenotypic differences provided by the reviewer cannot be fully excluded, since we indeed observed increased fibroblast frequency in Act mice at 3- and 5-days post-injury (see Fig 2b), although the increase was only approximately 1.5-fold. These additional fibroblasts may indeed contribute to the observed wound healing phenotype, especially at the early stage of repair. However, the increase in fibroblast numbers did not persist in activin-overexpressing mice up to day 7 post-injury, when collagen maturation and scar formation begin in earnest. It is after day 7 that we observed major differences between the genotypes at the histological (collagen maturation, asporin/periostin deposition), biochemical and mechanical levels. We have now clarified this issue in the discussion (pages 21 and 22).

For the RNA-seq analysis, we sorted approximately the same numbers of wound fibroblasts from mice of both genotypes and the total RNA concentration was normalized between all samples. Therefore, the differences in gene expression cannot be explained by a different number of fibroblasts. Furthermore, a comparison of the most strongly expressed genes in wound fibroblasts with activin-regulated or non-regulated genes shows little overlap, again pointing to a minimal contribution of increased fibroblast number to our identified activin-regulated gene signature (see Venn diagram Supplementary Fig. 3b *iii*).

Most importantly, we performed additional *in vitro* experiments to confirm the direct effect of activin A on fibroblasts, especially when it comes to the most significantly activin-regulated genes (*Postn* and *Aspn*) identified by RNA-seq:

- a.) We tested if some of the genes that are regulated by activin A *in vivo* are direct targets of activin signaling. Indeed, we identified multiple Smad2/3 binding elements in the promoter

or in the first intron of the *Postn* and *Aspn* genes (selected as examples for activin targets), of which some are highly conserved among different species (see Supplementary Fig. 4b). Chromatin immunoprecipitation (ChIP) experiments demonstrate that Smad2/3 indeed bind to these elements in response to activin A treatment, thus identifying *Aspn* and *Postn* as direct activin targets. We used immortalized fibroblasts from young mice for this purpose, since we need a large number of cells for these experiments and primary fibroblasts from adult mice grow poorly and can only be passaged once. These new results are now shown in Fig. 4c.

- b.) We established fibroblasts from 5-day wounds, starved them to reduce the effect of the TGF- β in the medium, and treated them with recombinant activin A. qRT-PCR experiments revealed that recombinant activin A also induces expression of *Inhba* and *Wisp1* in wound fibroblasts. Interestingly, expression levels of *Col1a1* and *Aspn* were already significantly higher in cultured wound fibroblasts of Act vs Wt mice, and recombinant activin A did not further induce their expression at the time point that we analyzed. These findings, shown in Fig. 4b, suggest that the wound fibroblasts from Act mice are already activated and that resulting epigenetic alterations may be maintained *in vitro*. Indeed, expression of *Col1a1* and *Lox* was significantly higher in fibroblasts from 5-day wounds compared to fibroblasts from non-injured skin of mice at the same age. These new results are now shown in Supplementary Fig. 4a.
- c.) We generated immortalized human keratinocyte lines (HaCaT cells) with doxycycline (Dox)-inducible activin A overexpression and used them in 2D co-cultures with immortalized fibroblasts with Dox-inducible expression of a dominant-negative activin receptor IB mutant (dnActRIB). FACS isolation of keratinocytes and fibroblasts from these co-cultures and subsequent qRT-PCR analysis revealed that activin A overexpression in keratinocytes promotes the expression of *Aspn* in fibroblasts of the co-culture. Importantly, the increase in *Aspn* expression was abolished when fibroblasts expressing dnActRIB were used for the co-culture. *Lox* and *Postn* expression was also strongly suppressed in fibroblasts of dnActRIB mice, indicating that their expression is dependent on activin receptor signaling. These new data are shown in Fig. 4f. Furthermore, we show by immunofluorescence analysis that activin A promotes the deposition of collagen type I, fibronectin, asporin and periostin in the ECM of these co-cultures, in particular in the vicinity of the activin-overexpressing keratinocytes. These effects were again abrogated when keratinocytes were co-cultured with fibroblasts expressing dnActRIB. These new results are now shown in Fig. 4e and Supplementary Fig. 4c.

We believe that our new data clearly demonstrate that activin A acts directly on fibroblasts and changes their transcriptional signature and matrisome, which can be considered as 'reprogramming'. However, we agree that this term may be too strong and we have therefore modified the title as follows: "Activin-mediated alterations of the fibroblast transcriptome and matrisome control the biomechanical properties of skin wounds". Furthermore, we toned down our conclusions regarding the transcriptomics data of Act vs WT mice obtained with the bulk fibroblast population (see 1st paragraph of page 9 and last paragraph of page 12).

Other points:

1. *The measurement of wound strain is interesting and potentially very relevant, but seems to be based on manual pulling of medical tape, which could lead to precision problems due to variable and non-controlled loading rates of the strain. It would be important to see a calibration and*

reproducibility curves of this method for example using sheets of viscoelastic materials of known properties.

Our reply: We thank the reviewer for this important comment, which concerns the influence of the applied loading rate on the elastograms produced through our *in vivo* technique to monitor healing progression, as well as on the corresponding quantification of wound deformability. The mechanical behavior of skin is characterized by a viscoelastic and poroelastic response, as reported by us (Barbarino et al., 2011, PMID 21362059; Weickenmeier et al., 2015, PMID 26584965; Pensalfini et al., 2018, PMID 29149656; Wahlsten et al., 2019, PMID 30806838) and others (Diridollou et al., 1998, PMID 9550180; Luebberding et al., 2013, PMID 23889488; Crichton et al., 2011, PMID 21458062; Del Prete et al., 2004, PMID 15336923; Wang et al., 2015, PMID 25803703), and therefore its resistance to deformation is time- and deformation rate-dependent, as for many other soft biological tissues. The influence of rate-dependent mechanical behavior on the repeatability of freehand elastography techniques has been previously analyzed in ultrasound-based settings using tissue phantoms (see e.g., Bamber et al., 2013, PMID 23558397, Doyley et al., 2001, PMID 11731048; Havre et al., 2008, PMID 18524458). Those studies observed an effect of the strain rate on the applied level of deformation and its reproducibility. However, to the best of our knowledge, no significant influence was reported on the ability of elastography to detect strain differences between tissue regions characterized by different mechanical properties (e.g. shear modulus). We expect such an influence to be present in our measurements, although not to an extent that may substantially alter the identification of a low-strain region within the wound or the strain magnitude in such a region.

To corroborate the above statement, we first determined whether important strain rate variations might have occurred in the *in vivo* experiments presented in our manuscript. To this end, we identified the time point corresponding to an applied global strain of 10%, which was set a priori as a common strain level for the analysis of our elastograms, and used the frequency of image acquisition from the camera to quantify the rate at which the imposed deformation was applied. This analysis, performed on a total of 154 wounds, showed a modest variation of the imposed mean strain rate, with an average value of 9.4%/s and a standard error of the mean (SEM) of 0.2%/s (range: 4.4-23.0%/s). These data are now reported in the manuscript (second paragraph of page 19, and first paragraph of page 37).

To further analyze the robustness of our procedure, we evaluated whether the global strain value that was selected to assess the level of deformability of the wounds might affect the outcome of the measurement. Interestingly, we observed that repeating the analysis reported in Fig. 7e-f of the manuscript at a global strain of 5% instead of 10% did not significantly affect the relative deformability of the wound. This is shown by the comparison of the ratio between wound and global strains for the two selected global strain levels (shown in Supplementary Fig. 8a). Depending on the image quality, performing the analysis at a lower global deformation level might affect the reliability of the image-based local strain quantification. Nonetheless, for the setting adopted in our manuscript, equivalent results could be obtained with much lower skin stretch levels. This might constitute an advantage if the method is applied on particularly fragile tissues.

An additional argument in support of the modest variability of the proposed method is provided by the data shown in Supplementary Fig. 8h, where we show the quantification of the difference between independent sets of measurements performed on wounds of the same phenotype and at corresponding time points (days 3, 10 and 14). The differences are consistently below 2-3% of local strain (data reported at 90% confidence interval). In comparison with the typical changes in

wound deformability observed here, these data indicate that the method's reproducibility is satisfactory.

In response to the reviewer's comment, we additionally prepared synthetic model systems representing cutaneous wounds by inserting a 6 mm diameter disc of sandpaper between two sheets of acrylic elastomers, whose time-dependent mechanical characteristics have been previously quantified (see e.g. Wissler and Mazza, 2007). Following surface preparation for optical strain tracking (Hopf et al., 2016, PMID 26990071), the specimen (clamp to clamp distance: 40 mm, width: 50 mm) containing the stiff inclusion at its center was mounted on our previously-presented custom-made setup for mechanical testing (Hopf et al., 2016, PMID 26990071) and elongated to about 15% nominal strain at a strain rate of either 10%/s or 0.1%/s (i.e. different by 2 orders of magnitude). The former value was chosen to be representative of the strain rate applied in the *in vivo* experiments, whereas the latter was chosen to represent a particularly slow elongation. As a consequence of the viscoelastic behavior of the elastomer, the force required for the low strain rate extension was about half of that corresponding to the faster loading (Supplementary Fig. 8b). As can be seen from Supplementary Fig. 8c, the elastograms obtained using the algorithm described in our manuscript display only modest differences between the two cases, despite the large strain rate variation applied. Quantifying the strain in the center of the stiff inclusion (Supplementary Fig. 8d) further shows a limited increase (approximately 2.2% vs 1.7%) despite the extremely large, namely 100-fold, variation in the imposed strain rate. These data support the usefulness of a manual pulling technique.

Finally, we corroborated this experimental evidence with numerical simulations of load application to wounded skin, and quantified the influence of strain rate also in this *in silico* setting. To this end, we modeled a 1 mm-thick skin specimen with planar dimensions of 25x25 mm² and including a 5 mm diameter wound at its center; elongating the specimen by 10% in either 100 s or 1 s yielded again strain rates differing by a factor of 100. The constitutive behavior of healthy skin was directly adopted from previous studies (Weickenmeier et al., 2015, PMID 26584965; Weickenmeier and Jabareen, 2014, PMID 24421263), where a model of skin's time-dependent mechanical behavior *in vivo* was derived and validated; the parameter μ_0 , akin to a shear modulus for the tissue, was increased by one order of magnitude in order to represent the wounded tissue behavior. As shown in Supplementary Fig. 8e, the faster loading rate yielded an almost two-fold increase in the reaction force, which is in line with the experimental results shown in Supplementary Fig. 8b on VHB 4910TM. Notably, despite the important variation in the imposed strain rate, the predicted strain patterns obtained from finite-element simulations are not visibly different inside or outside of the stiff inclusion (Supplementary Fig. 8f). Furthermore, quantifying the strain level at the center of the wounded region confirmed once again the modest influence of strain rate on this parameter (Supplementary Fig. 8g).

Taken together, these results demonstrate that the manual method for strain application adopted here does not constitute a significant disadvantage of the proposed technique for longitudinal *in vivo* monitoring of wound deformability. Indeed, while we have shown that the relative strain level within the wound might be affected by the rate of the imposed deformation, this influence was found to be quite modest in the investigated range of strain rates (which seems reasonable for the envisaged applications of the method). Most importantly, the observed influence is much lower than the relative stiffness variations observed in the course of healing (Fig. 7f). Finally, it should be mentioned that manual control of the imposed deformation is beneficial in terms of safety of the proposed technique, in that it offers direct haptic feedback against possible tissue overloading and thus potential wound damage.

Reviewers' Comments:

Reviewer #1:

Remarks to the Author:

The authors have comprehensively revised and increased the mechanistic insight of their work. Several substantial new experiments have been added that now address my main concerns. My other concerns, that did not require new experiments, are sufficiently clarified. As the result of these revisions, I believe the manuscript is much improved.

Reviewer #2:

Remarks to the Author:

The authors have made a significant effort in revising the manuscript and in particular the new in vitro experiments strengthen their conclusions. The reviewer also appreciates the thorough effort to validate the measurements of tissue mechanics.

However, I still maintain my critical view on the transcriptomic analyses, even taken to consideration the PCA plots in Supplementary 2. The fact is that these samples are still very similar to each other and the variation between individual samples is larger than that between genotypes. Additionally, the biological effect is very small, highlighted by Fig. 3d, where half of the genes listed show only a 10% or less change in their expression, and the biological meaningfulness of these marginal changes is not clear. Having said that, I appreciate that the authors have toned down their conclusions and also changed the title, which is now more appropriate. I would still recommend toning this down, eg. by rewording/removing the added sentence in the last paragraph of p7 to more clearly indicate that variability between samples is similar than that of between genotypes.

A few points should be addressed prior to publication

1. It is not clear to this reviewer how the experiment in Fig 4b demonstrates a positive feedback of Activin. First, the differences in mRNA are marginal and 2 out of 4 cases activin does not have a differential effect in the overexpresses vs wild types as would be expected. This should be explained and the conclusions should be toned down.
2. ChIP in Fig. 4 needs to be analyzed by quantitative real time PCR and quantified in 3 independent replicates

REVIEWERS' COMMENTS:

Reviewer #1 (Remarks to the Author):

The authors have comprehensively revised and increased the mechanistic insight of their work. Several substantial new experiments have been added that now address my main concerns. My other concerns, that did not require new experiments, are sufficiently clarified. As the result of these revisions, I believe the manuscript is much improved.

Our reply: We thank the reviewer for the positive comments and for the very important suggestions/criticisms to the first version, which helped a lot to improve the manuscript.

--

Reviewer #2 (Remarks to the Author):

The authors have made a significant effort in revising the manuscript and in particular the new in vitro experiments strengthen their conclusions. The reviewer also appreciates the thorough effort to validate the measurements of tissue mechanics.

Our reply: We thank the reviewer for the positive comments and for the very important suggestions/criticisms to the first version, which helped a lot to improve the manuscript.

However, I still maintain my critical view on the transcriptomic analyses, even taken to consideration the PCA plots in Supplementary 2. The fact is that these samples are still very similar to each other and the variation between individual samples is larger than that between genotypes. Additionally, the biological effect is very small, highlighted by Fig. 3d, where half of the genes listed show only a 10% or less change in their expression, and the biological meaningfulness of these marginal changes is not clear. Having said that, I appreciate that the authors have toned down their conclusions and also changed the title, which is now more appropriate. I would still recommend toning this down, eg. by rewording/removing the added sentence in the last paragraph of p7 to more clearly indicate that variability between samples is similar than that of between genotypes.

Our reply: We agree that the differences between genotypes are small, but as mentioned in the manuscript, they are underestimated by the analysis of all fibroblasts. We have further clarified this in the text (page 7-8). Most importantly, relevant changes were validated using various *in vitro* and *in vivo* analyses. Nevertheless, we reworded the added sentence in the last paragraph of p7 as requested.

A few points should be addressed prior to publication

1. It is not clear to this reviewer how the experiment in Fig 4b demonstrates a positive feedback of Activin. First, the differences in mRNA are marginal and 2 out of 4 cases activin does not have a differential effect in the overexpresses vs wild types as would be expected. This should be explained and the conclusions should be toned down.

Our reply: This is correct. *INHBA* is a primary response gene and the mRNA half-life is approximately 30 min (PMID 27001469). Therefore, it is not surprising that the mRNA levels in the fibroblasts of the transgenic mice are only marginally increased when the cells are cultured and thus removed from the source of activin. We agree that the further increase is also rather small in this experiment. Therefore, we have removed these data, in particular since we previously showed a more convincing *INHBA* autoinduction in fibroblasts, which was published in another context. Therefore, we refer to this manuscript (Ref. 28) on page 8, first paragraph.

2. ChIP in Fig. 4 needs to be analyzed by quantitative real time PCR and quantified in 3 independent replicates

Our reply: The data were reproduced in two additional independent experiment and a repetition experiment is shown in Fig. S4c.

There is a clear difference between activin-treated cells and vehicle-treated cells and the binding of SMAD2/3 to the SBE in response to activin A was obvious in all experiments. We used gels in this case to demonstrate the correct size of the DNA amplification product. We would of course repeat the experiment and use real-time PCR to quantify the changes. Unfortunately, we had a complete shut-down of our lab three weeks ago because of Covid-19 and we will not be allowed to perform any experiments at least until the end of May. Therefore, we sincerely hope that the data we provide (and which were reproduced in different independent experiments) are sufficient for publication. Otherwise publication would be strongly delayed and this would be problematic, because the topic is competitive.